


# Identification of linear response functions from arbitrary perturbation experiments in the presence of noise −

## Part II. Application to the land carbon cycle in the MPI Earth System Model

Guilherme L. Torres Mendonça[1,2], Julia Pongratz[2,3], and Christian H. Reick[2]

[1]International Max Planck Research School on Earth System Modelling, Hamburg, Germany
[2]Max Planck Institute for Meteorology, Hamburg, Germany
[3]Ludwig-Maxmillians-Universität München, Munich, Germany

**Correspondence:** Guilherme L. Torres Mendonça (guilherme.mendonca@mpimet.mpg.de)

**Abstract.** The Response Function Identification method introduced in the first part of this study is applied here to investigate the land carbon cycle in the Max Planck Institute for Meteorology Earth System Model. We identify from standard $C^4MIP$ 1% experiments the linear response functions that generalize the land carbon sensitivities $\beta$ and $\gamma$. The identification of these generalized sensitivities is shown to be robust by demonstrating their predictive power when applied to experiments not used

for their identification. The linear regime for which the generalized framework is valid is estimated, and approaches to improve the quality of the results are proposed. For the generalized $\gamma$-sensitivity, the response is found to be linear for temperature perturbations until at least 6 K. When this sensitivity is identified from a $2\times CO_2$ experiment instead of the 1% experiment, its predictive power improves, indicating an enhancement in the quality of the identification. For the generalized $\beta$-sensitivity, the linear regime is found to extend up to $CO_2$ perturbations of 100 ppm. We find that nonlinearities in the $\beta$-response arise mainly

from the nonlinear relationship between Net Primary Production and $CO_2$. By taking instead of $CO_2$ the resulting Net Primary Production as forcing, the response is approximately linear until $CO_2$ perturbations of about 850 ppm. Taking Net Primary Production as forcing also substantially improves the spectral resolution of the generalized $\beta$-sensitivity. For the best recovery of this sensitivity, we find a spectrum of internal time scales with two peaks, at 4 and 100 years. Robustness of this result is demonstrated by two independent tests. We find that the two-peak spectrum can be explained by the different characteristic

time scales of functionally different elements of the land carbon cycle. The peak at 4 years results from the collective response of carbon pools whose dynamics is governed by fast processes, namely pools representing living vegetation tissues (leaves, fine roots, sugars, and starches) and associated litter. The peak at 100 years results from the collective response of pools whose dynamics is determined by slow processes, namely the pools that represent the wood in stem and coarse roots, the associated litter, and the soil carbon (humus). Analysis of the response functions that characterize these two groups of pools shows that

the pools with fast dynamics dominate the land carbon response only for times below 2 years. For times above 25 years the response is completely determined by the pools with slow dynamics. From 100 years onwards only the humus pool contributes to the land carbon response.



## 1 Introduction

In Part I of this study we developed a method to identify linear response functions from arbitrary perturbation experiments. The RFI method (Response Function Identification method) was tested by means of artificial toy model simulations. Here, we demonstrate the applicability of our method to a practical problem: We investigate the dynamics of the land carbon cycle in the Max Planck Institute for Meteorology Earth System Model (MPI-ESM; see Appendix A). In particular, we show how our RFI method provides insight into two aspects of central relevance to carbon-cycle research: The sensitivity of the land carbon cycle to changes in atmospheric $CO_2$ and its distribution of internal time scales.

The global carbon cycle plays a critical role in mitigating climatic effects from $CO_2$ emissions. According to the yearly published Global Carbon Budget (Friedlingstein et al., 2020), about half of the anthropogenic $CO_2$ emitted from pre-industrial times to 2019 has been taken up from the atmosphere by the ocean and the terrestrial biosphere. Despite its relevance to climate, the dynamics of the carbon cycle is still poorly understood (Ilyina and Friedlingstein, 2016). Improving our understanding of this dynamics, in particular in response to $CO_2$ perturbations, is therefore one of the major challenges of climate research (Marotzke et al., 2017).

The most advanced tools used to study the response of the carbon cycle to perturbations and its effect on climate are known as Earth System Models, which are complex numerical models that simulate the interaction between climate and the carbon cycle. To systematically investigate this interaction, for more than a decade now internationally coordinated simulation exercises have been performed with several Earth System Models within the Coupled Climate Carbon Cycle Model Intercomparison Project (C[4]MIP; see Fung et al., 2000; Friedlingstein et al., 2006) that today belongs to the international Coupled Model Intercomparison Project (CMIP; see Taylor et al., 2012). Results from such exercises show a general agreement across models in the ocean carbon response, while the response of land carbon presents a large model spread (Friedlingstein et al., 2006; Arora et al., 2013, 2019).

To better understand the reasons for this spread one may follow a top-down strategy by investigating which processes dominate the various contributions to the global response. A typical approach in this direction is to consider two global-scale contributions: One arising from the sensitivity of the land carbon cycle to the radiative effect of $CO_2$ acting via greenhouse warming, and another from its sensitivity to the biochemical effect of $CO_2$ concentrations on photosynthetic carbon assimilation (Arora et al., 2013; Schwinger et al., 2014; Adloff et al., 2018; Arora et al., 2019). The magnitude of each of these sensitivities is quantified by the $\gamma$ and $\beta$ values introduced by Friedlingstein et al. (2003). But although these values give insight into the magnitude of the sensitivities, they cannot be seen as properties of the land carbon cycle alone. The reason is that $\gamma$ and $\beta$ quantify the sensitivity of land carbon to $CO_2$ perturbations only for a particular perturbation scenario, so that for different scenarios one may obtain different values (Gregory et al., 2009; Arora et al., 2013).

To quantify this sensitivity in a more systematic way and thereby gain deeper insight into the land carbon dynamics one needs a more general formalism. For small changes in atmospheric $CO_2$ one can shown that by accounting for the memory of the carbon cycle these values are generalized to linear response functions, which in turn can be seen as properties of the land carbon cycle itself (Rubino et al., 2016; Enting and Clisby, 2019, see also Appendix B). As a result, these linear response





functions characterize the land carbon sensitivities for any perturbation scenario. For this reason these functions will here be called land carbon *generalized sensitivities*.

The essential step to investigate the carbon-cycle dynamics within this general formalism is the identification of the gen-
eralized sensitivities. Typically, the $\gamma$ and $\beta$ values proposed by Friedlingstein et al. (2003) are obtained taking data from standardized C[4]MIP simulation experiments where, starting from an equilibrium state, atmospheric $CO_2$ concentration is increased by 1% each year. Since data from such 1% experiments performed with several Earth System Models are readily available in international databases, one would be interested in identifying the generalized $\gamma$- and $\beta$-sensitivities as well from these experiments. But methods in the literature to identify response functions from data require special perturbation exper-
iments, and C[4]MIP experiments were not tailored for this purpose. It is here that our RFI method is useful: Because it was designed to derive response functions from experiments driven by any arbitrary perturbation, in the present study we show that by this method one can robustly derive the land carbon generalized sensitivities for the MPI-ESM taking data from standard C[4]MIP 1% experiments. To make sure the identified generalized sensitivities are indeed characteristics of the land carbon cycle in the MPI-ESM, we demonstrate their predictive power by applying them to predict the response of the model in several
experiments that were not used for their identification. In preparation for future studies applying these generalized sensitivities to study the dynamics of the carbon-climate system in C[4]MIP models, we also investigate various ideas to improve the quality of the recovery of the response functions by using additional types of data routinely available in C[4]MIP simulations or using log-transformed data to account at least partially for process-immanent nonlinearities that hinder the usage of experiment data from larger levels of forcing.

Apart from giving a systematic quantification of the sensitivities, linear response functions can be a powerful tool to gain insight into the internal dynamics of the carbon cycle. Because response functions fully characterize the linear response of a system, they contain information on its distribution of internal time scales, i.e. the weights with which characteristic time scales from internal processes contribute to the macroscopic response of the system. These weights may shed light into which are the most relevant processes to the response at different time scales. For our application to the land carbon, such information may
give hints into the main processes influencing the model spread found in the C[4]MIP results.

But while several studies have tried to obtain the weights of different time scales in the carbon cycle by fitting response functions to a sum of few exponents (Maier-Reimer and Hasselmann, 1987; Enting and Mansbridge, 1987; Enting, 1990; Joos et al., 1996; Pongratz et al., 2011; Joos et al., 2013; Colbourn et al., 2015; Lord et al., 2016), in principle it is not clear to what extent such results can be trusted, let alone interpreted. The reason is that finding these weights from data is a severely
ill-posed problem that requires special methods to be dealt with (Istratov and Vyvenko, 1999). In contrast to the classical fitting procedures, by employing a regularization technique (Groetsch, 1984; Engl et al., 1996) combined with a novel estimation of the noise level in the data (see Part I) our RFI method accounts for this ill-posedness. In addition, instead of assuming that the response functions result from only few time scales, the RFI method recovers a continuous spectrum of time scales, in agreement with what one would expect when studying the carbon cycle response (Forney and Rothman, 2012a). In the present
work we show that, in contrast to results obtained with classical fitting procedures, spectra recovered by the RFI method may be *reliable* and even *interpretable*. For this purpose, we investigate a relatively detailed spectrum of time scales that arises from





a high-quality recovery of the generalized $\beta$-sensitivity. We examine (i) the robustness of the obtained spectrum and (ii) the explanation for its time-scale structure.

An additional novelty introduced here is a simple procedure to estimate the linear regime of the response, i.e. the range of
perturbation strengths for which the response of the system can be considered linear. As discussed in Part I, the presence of traces of nonlinear responses in the data can severely deteriorate the recovery of the response function. Hence, to make sure that the data from which the response function is recovered contain no strong nonlinearities, one must be able to estimate the linear regime of the response. Because the response functions will be derived from 1% experiments, we introduce a technique to estimate with the aid of additional simulations the linear regime from this type of experiment. By this technique the linear
regime of the response of land carbon to changes in $CO_2$ and climate for the MPI-ESM will be estimated.

The outline of the paper is as follows. In the next section we introduce the methodology of the study including the RFI method, the $C^4MIP$ experiments, and the technique to estimate the linear regime of the response from "percent" experiments. In sections 3 and 4 we identify and investigate the generalization of the $\gamma$- and $\beta$-sensitivities in the MPI-ESM. In section 5 we investigate the detailed spectrum of time scales obtained for the generalized $\beta$-sensitivity. In section 6 the results are
summarized and discussed.

## 2   Methodology

In this section we introduce the methodology employed throughout the study. We start by briefly introducing the RFI method (for a detailed description please refer to Part I), the $C^4MIP$-type experiments considered here, and technical details for the identification of the generalized sensitivities. Then, we present our procedure to estimate the linear regime of the response from
"percent" experiments.

### 2.1   RFI method and $C^4MIP$ experiments

The RFI method identifies the response function $\chi(t)$ taking data from the response $Y(t)$ – in our example the global land carbon – and the perturbation $f(t)$ – atmospheric $CO_2$ or temperature (see below) – assuming the following ansatz based on linear response theory (see Part I):

$$\Delta Y(t) = \int_0^t \chi(t-s)\Delta f(s)ds + \eta(t), \tag{1}$$

where $\eta(t)$ is a noise term. Here $\Delta Y(t)$ and $\Delta f(t)$ mean that we are taking only the change in the variables with respect to their equilibrium values from a control simulation. Following Forney and Rothman (2012a), the spectrum of internal time scales is obtained by assuming that the response function $\chi(t)$ can be represented by an overlay of exponential modes:

$$\chi(t) = \int_0^\infty g(\tau)e^{-t/\tau}d\tau. \tag{2}$$





To account for the large range of time scales in the carbon cycle (Ciais et al., 2013) it is useful to rewrite Eq. (2) in terms of $\log_{10} \tau$, so that

$$\chi(t) = \int_{-\infty}^{\infty} q(\tau) e^{-t/\tau} d\log_{10}\tau, \quad \text{with} \quad q(\tau) := \tau \ln(10) g(\tau). \tag{3}$$

The spectrum of time scales is then given by $q(\tau)$, which following the terminology from Part I we call simply *spectrum*. The problem is solved by discretizing Eq. (1) and Eq. (3), prescribing a distribution of time scales $\tau$, taking the data on $\Delta Y(t)$ and $\Delta f(t)$, and solving a minimization procedure for the spectrum $q(\tau)$. The parameters to prescribe the distribution of time scales are taken identically to those chosen for the application to the toy model in Part I. To treat the ill-posedness we employ Tikhonov-Phillips regularization (Phillips, 1962; Tikhonov, 1963) in a Singular Value Decomposition (SVD) framework that gives the solution by the expansion (Hansen, 2010; Bertero, 1989)

$$\boldsymbol{q}_\lambda = \sum_{i=0}^{M-1} f_i(\lambda) \frac{\boldsymbol{u}_i \bullet \boldsymbol{\Delta Y}}{\sigma_i} \boldsymbol{v}_i, \tag{4}$$

where $M$ is the number of time scales, $\boldsymbol{u}_i$ and $\boldsymbol{v}_i$ are the singular vectors, $\sigma_i$ are the singular values, $\lambda$ is the regularization parameter, and $f_i(\lambda)$ are the filter functions

$$f_i(\lambda) = \frac{\sigma_i^2}{\sigma_i^2 + \lambda}. \tag{5}$$

The regularization parameter $\lambda$ is determined by the discrepancy method (Morozov, 1966) with noise level estimated from a SVD analysis of the data combined with information from the control simulation. Once the spectrum $\boldsymbol{q}_\lambda$ is found, the response function $\chi(t)$ follows from Eq. (3).

All linear response functions are identified by the RFI method taking data from C$^4$MIP-type experiments performed with the MPI-ESM. We focus on identifying the response functions from standard 1% experiments that are widely available in international databases. In addition, to examine the quality of the results, we identify as well some response functions taking data from additional experiments. To investigate the robustness of the identified response functions, we employ them to predict the response of the MPI-ESM in several experiments not used for the identification. A summary of the experiments considered in the study is given in Table 1, with forcing scenarios shown in Fig. 1.

The variables taken for the identification of the response functions are the ones relevant for the quantification of the land carbon sensitivities $\gamma$ and $\beta$, respectively:

(a) The change in global land carbon in response to changes in global land temperature;

(b) The change in global land carbon in response to changes in atmospheric $CO_2$.

Global land carbon is computed as the sum of the total land carbon over all grid cells of the model. Global land temperature is calculated as the mean near-surface air temperature over land at 2 m height. The changes are computed as $\Delta Y(t) = Y(t) - Y_{PI}$, with $Y_{PI}$ being the mean value of observable $Y$ from a control simulation at pre-industrial conditions. Since the main interest lies in long-term variations, annual mean data are used.





As demonstrated in Appendix C, the generalization of the $\beta$-sensitivity can be shown to be monotonic. Therefore, in the following we will derive it employing the additional noise level adjustment in the RFI algorithm (step 6 of Fig. 1 in Part I). Since the generalization of the $\gamma$-sensitivity is not known to be monotonic, for this sensitivity the RFI algorithm will be applied without the additional adjustment.

**Table 1.** $C^4$MIP-type experiments considered in this study. Forcings are shown in Fig. 1. Acronyms "rad" and "bgc" refer to standard CMIP model setups used to calculate the climate-carbon cycle sensitivities. In the "rad" (radiatively coupled) setup only the radiation code of the model is affected by changes in atmospheric $CO_2$. This setup is used to calculate $\gamma$. In the "bgc" (biogeochemically coupled) setup only the biogeochemistry of the model is affected by changes in atmospheric $CO_2$. This setup is used to calculate $\beta$. In brackets are names of standard CMIP experiments.

| Type | Forcing | Description |
|---|---|---|
| Percent | 0.5% rad | |
| | 0.5% bgc | |
| | 0.75% rad | |
| | 0.75% bgc | |
| | 1% rad (esmFdbk1) | $CO_2$ is increased from its pre-industrial value at the |
| | 1% bgc (esmFixClim1) | specified percent rate per year. |
| | 1.5% rad | |
| | 1.5% bgc | |
| | 2% rad | |
| | 2% bgc | |
| Step | $1.1 \times CO_2$ rad | |
| | $1.1 \times CO_2$ bgc | $CO_2$ is abruptly increased from its pre-industrial value by |
| | $2 \times CO_2$ rad | the specified factor. |
| | $2 \times CO_2$ bgc | |
| Control | pre-industrial (piControl) | $CO_2$ is held fixed at its pre-industrial value. |

## 155 2.2 Estimating the linear regime from "percent" experiments

As described in Part I, the recovery of response functions is cursed by the presence of noise and nonlinearities. The RFI method is designed to cope with the former. In the present section we present a technique to cope with the latter, but at the expense of performing additional response experiments. This technique will serve as a complement to the RFI method in the application to the land carbon cycle in the following sections. By these additional experiments we will determine the range of forcing


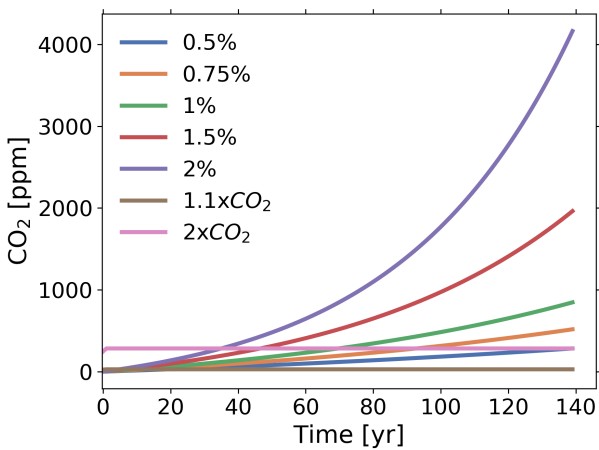

**Figure 1.** Forcings for the C$^4$MIP-type experiments considered in this study.

strengths for which the response can be considered linear – an analysis in general not possible using only the control and perturbation experiments on which the RFI method is based.

To introduce this technique we use the example of simulations with the linear toy model employed in Part I. To demonstrate the effect of nonlinearities on the recovery of $\chi(t)$, following Part I we artificially add a nonlinear term $-aY^2(t)$ to the linear response $Y(t)$ of the toy model:

$$Y_{nonlin}(t) := Y(t) - aY^2(t). \tag{6}$$

In this way we obtain a nonlinear response $Y_{nonlin}(t)$, with nonlinearity strength controlled by the parameter $a$, from which $\chi(t)$ is derived. In addition to including this nonlinear term in the toy model response, to introduce the technique we will need to quantify the quality of the recovery of the response function. Since in our application to the land carbon the "true" response function is not known a priori, following Part I we quantify the quality of the recovery indirectly by measuring the quality with

which the recovered response function can predict the response of the model in experiments not used for the recovery itself. For this purpose we introduce the *prediction error*

$$\varepsilon_k := \frac{||\boldsymbol{\Delta Y}^k - \boldsymbol{\chi} \star \boldsymbol{\Delta f}^k||}{||\boldsymbol{\Delta Y}^k||}, \tag{7}$$

where $\star$ stands for the convolution operation, $\boldsymbol{\Delta Y}^k$ and $\boldsymbol{\Delta f}^k$ are the response and the perturbation in experiment "k", and $\boldsymbol{\chi}$ is the response function recovered from the 1% experiment.

We can now present our technique. Taking first a purely linear situation ($a = 0$) we show in Fig. 2(a) the prediction error (7) when using the response function obtained from a 1% simulation to predict the response from two other %-simulations with smaller growth rate. More precisely, performing a sequence of 1% experiments for increasingly longer simulation periods, we calculated for each experiment the response function and used it to predict the response for a 0.5% and 0.75% experiment

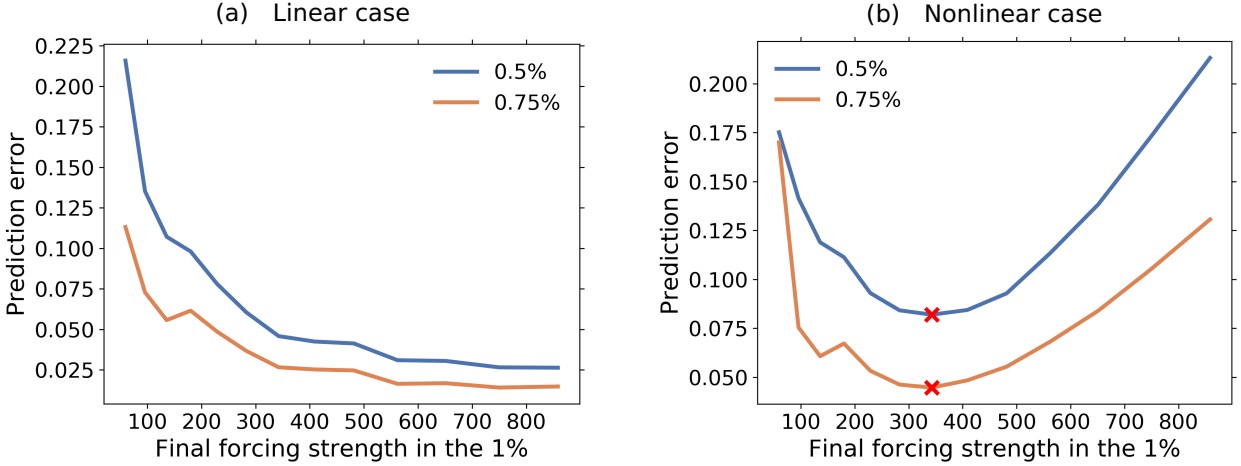

**Figure 2.** Toy model example for the identification of the linear regime by using additional experiments. Shown is the prediction error (7) for the response of 0.5% and 0.75% experiments as obtained from the response function calculated by the RFI method from 1% experiments. The prediction errors are plotted against the final forcing strengths of a sequence of 1% experiments with increasing time series length. The crosses at the minima indicate the final forcing strength for which the response function is optimally recovered (see text). Subfigure (a) shows the behaviour for the fully linear toy model ($a = 0$) and subfigure (b) the behaviour in the presence of nonlinear contributions to the response ($a = 5 \times 10^{-5}$). For the purpose of demonstrating more clearly the increase in the prediction error for a decrease in the forcing strength, we include in the plot cases where the forcing strength is extremely small, corresponding to very small time series lengths. To deal with such cases, we set for a number of data points $N < 30$ the number of time scales $M = N$ (see parameters for the RFI method in Part I). For such small number of time scales, usually no plateau in the singular values spectrum is found (step 2 of Fig. 1 in Part I). Therefore, for these special cases we also modify the algorithm to interpret the two smallest singular values as a plateau, since their small magnitude makes them have a similar effect to those singular values belonging to the plateau itself. In addition, to illustrate the most general case where $\chi(t)$ is not known to be monotonic, we exclude here the monotonicity check (step 6 of Fig. 1 in Part I).

covering the same simulation period. Then we plotted in Fig. 2(a) the prediction error against the final forcing strength of the
1% experiment. As a result, it is seen that with increasing final forcing strength the prediction error decreases. This happens because in this linear case the SNR is increasing with increasing simulation period, i.e. with increasing final forcing strength, so that the recovery of the response function continuously improves.

Calculating the prediction error only for experiments with smaller growth rate gets important in the next case where non-linearities are considered (Fig. 2(b)). This plot was obtained by the same procedure except that we took for the nonlinearity
parameter a value $a > 0$. As seen, in this nonlinear case the prediction error is first improving but deteriorating afterwards. For small forcing, nonlinearities are small and therefore the prediction error behaves as in the linear case, i.e. it decreases with final forcing strength. But when forcing strengths get larger, nonlinearities start to contribute substantially to the response, thereby causing a deterioration of the recovery of the response function and consequently the prediction error once more increases. This increase of the prediction error can be unambiguously traced back to the presence of nonlinearities in the 1% simulation





because the prediction error was calculated only for experiments with smaller growth rate, i.e. smaller forcing strength throughout the whole simulation. Therefore, nonlinearities contribute already substantially to the response in the 1% simulation before they get relevant in the other experiments. Accordingly, with this type of experiment setup we can be sure that the increase in the prediction error comes solely from the deterioration of the recovery of the response function and not from nonlinearities in the additional experiments used for calculating the prediction error.

Obviously, for forcing strengths smaller than at the minimum, nonlinearities do not hinder the recovery of the response function so that one can consider this to be the regime of linear system behaviour. In view of the trade-off between noise and nonlinearities, for the 1% experiment the response function is thus optimally recovered when taking as final forcing strength the value at the minimum of the prediction error curve. Similarly, if the error curve has no minimum (as in the linear case shown in Fig. 2(a)) the optimal recovery is obtained from the experiment with the largest forcing strength.

With the presentation of this additional technique to identify the linear regime we are ready for the application to the MPI-ESM in the next sections.

## 3 Generalized sensitivity $\chi_\gamma$

In this section we identify from MPI-ESM simulations the linear response function $\chi_\gamma$ (generalization of the $\gamma$-sensitivity), defined by

$$\Delta C^{rad}(t) = \int_0^t \chi_\gamma(t-s)\Delta T(s)ds + \eta(t), \tag{8}$$

where now the response is $\Delta Y(t) := \Delta C^{rad}(t)$ and the forcing is $\Delta f(t) := \Delta T(t)$, with $\Delta C^{rad}(t)$ being the change in global land carbon obtained in the "rad" experiment (see Table 1) and $\Delta T(t)$ the change in global land temperature.

That $\chi_\gamma$ indeed generalizes $\gamma$ can be understood by considering that $\gamma$ is defined by

$$\gamma(t) = \frac{\Delta C^{rad}(t)}{\Delta T(t)} \overset{(8)}{=} \frac{1}{\Delta T(t)} \int_0^t \chi_\gamma(t-s)\Delta T(s)ds, \tag{9}$$

for a negligible noise $\eta(t)$. From Eq. (9) it is clear that by knowing $\chi_\gamma(t)$ one can compute the response $\Delta C^{rad}(t)$ and thereby $\gamma(t)$ for any time-dependent perturbation $\Delta T(t)$, as long as the perturbation strength is small. Hence, $\chi_\gamma(t)$ can be seen as a property of the land carbon system and a generalization of $\gamma(t)$.

### Estimating the linear regime

As a first step in obtaining a proper approximation of $\chi_\gamma(t)$, we investigate what maximum forcing strength can be used to
assure that the recovered $\chi_\gamma(t)$ is not spoiled by the presence of nonlinearities. Using the technique introduced in section 2.2, we show in Fig. 3 the prediction error (7) for $\chi_\gamma(t)$ recovered from the 1% rad experiment as a function of the final forcing strength in the 1% rad experiment. There is no clear minimum so that for the data available the recovery of $\chi_\gamma(t)$ seems not to be limited by nonlinearities. For optimal recovery we thus take the full time series, i.e. the maximum final forcing strength.


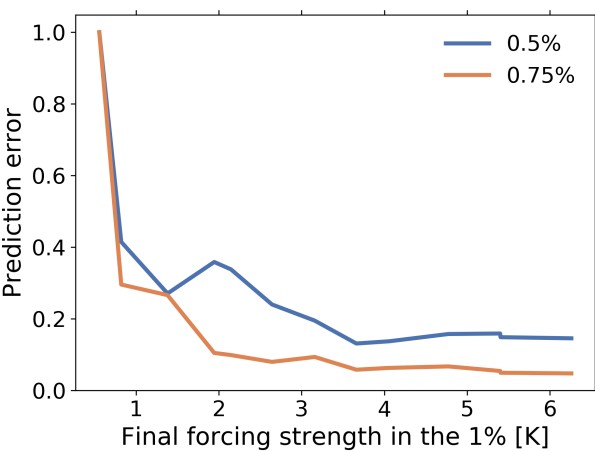

**Figure 3.** Prediction error (7) for the 0.5% and 0.75% rad experiments using $\chi_\gamma(t)$ recovered from the 1% rad experiment. The error is shown as function of the final forcing strength used for the recovery of $\chi_\gamma(t)$. No clear minimum is found so that the recovery seems not to be limited by nonlinearities.

## $\chi_\gamma$ and the quality of its recovery

The quality of the recovery can in principle be improved by taking an experiment with better SNR. To investigate if the recovery from the 1% rad experiment can be further improved, we applied the RFI algorithm also to recover $\chi_\gamma$ from the 2×CO$_2$ rad experiment. We chose the 2×CO$_2$ rad experiment because as shown in Fig. 4 it has smaller forcing strengths than the maximum forcing strength for the 1% rad experiment – therefore nonlinearities should also not limit the recovery – but is expected to carry useful information over a larger range of the response spectrum. This expectation can be justified as follows (MacMartin

and Kravitz, 2016). Taking the Laplace transform of Eq. (8) gives

$$\Delta \widetilde{C}^{rad}(p) = \widetilde{\chi}_\gamma(p)\Delta \widetilde{T}(p) + \widetilde{\eta}(p), \tag{10}$$

where the tilde denotes Laplace transformed functions. From Fig. 4 it is seen that for the 1% rad experiment the temperature $\Delta T$ behaves approximately as a linear function, which gives a Laplace transform $\Delta \widetilde{T}(p)$ proportional to $1/p^2$. For the 2×CO$_2$ rad experiment, the temperature behaves approximately as a step function (ignoring the transient in the first 20 years), which

gives a Laplace transform $\Delta \widetilde{T}(p)$ proportional to $1/p$. This means that for the same $\chi_\gamma$, the first term on the right-hand side of Eq. (10) – the "clean" response – decays to zero faster for the 1% rad experiment than for the 2×CO$_2$ rad experiment. Hence, assuming the same noise $\eta$ for both cases, the response from the 2×CO$_2$ rad experiment gets buried in the noise only at larger $p$, meaning that this experiment carries useful information until higher rate values $p$ than the response from the 1% rad experiment.

The response function $\chi_\gamma(t)$ recovered from the two types of experiments is shown in Fig. 5(a). As expected, the different experiments indeed result in different recoveries. Because we know from the analysis of Fig. 3 that nonlinearities do not limit


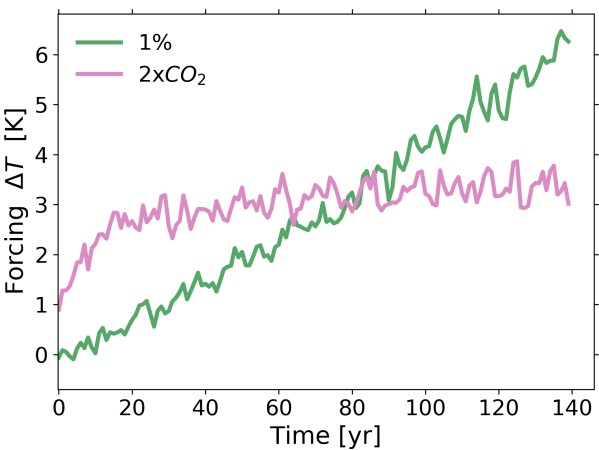

**Figure 4.** Forcing temperature $\Delta T(t)$ for 1% and $2\times CO_2$ rad experiments.

the recovery, the difference between the two response functions results probably from the difference in the quality of the data from the two types of experiments. To compare the robustness of each recovery, we analyze how well they predict the response from other experiments (Figs. 5(b) and (c)). If the response function is correctly recovered, it should be able to predict not only experiments with smaller but also experiments with higher forcing rate. Therefore, we include in the analysis also 1.5% and 2% rad experiments. To exclude errors that may be caused by nonlinearity, we take as a conservative estimate of the linear regime forcing strengths smaller than the final forcing strength at the end of the 1% rad experiment (which is approximately the maximum forcing strength; see temperature value at $t = 140$ years for the 1% rad experiment in Fig. 4). We take these values as an estimate of the linear regime because the 1% rad experiment has only 140 years so that no estimate for higher forcing strengths is available. Hence, for the 1.5% and 2% rad experiments the responses are expected to be reasonably predictable at least until the values marked with circles, where their respective forcing strengths reach this maximum forcing strength. All other experiments have forcing strengths smaller or equal to this maximum forcing strength so that they should be predictable for the whole time series.

Figure 5(b) shows the quality of the prediction using $\chi_\gamma(t)$ recovered from the 1% rad experiment. Visually, model response and prediction seem to have a comparable quality of agreement across the $1.1\times CO_2$, 0.5%, 0.75% and 1.5% rad experiments, while for the 2% and $2\times CO_2$ rad experiments there are larger discrepancies. For a quantitative analysis, we compute for the estimated linear regime the prediction error (7) for each experiment (right side of the plot). It is seen that the error varies from less than 10% for the 0.75% and 1.5% rad experiments to values between 10-20% for the 0.5%, 2% and $2\times CO_2$ rad experiments, and a significantly larger value of 57% for the $1.1\times CO_2$ rad experiment. To better understand these differences, it is important to note that as long as nonlinearities are small, experiments with higher forcing strength are expected to have smaller prediction error because they have higher SNR. This can be made plausible by considering that a perfectly recovered response function predicts a "clean" linear response (infinite SNR) with zero error, whereas the same response function can

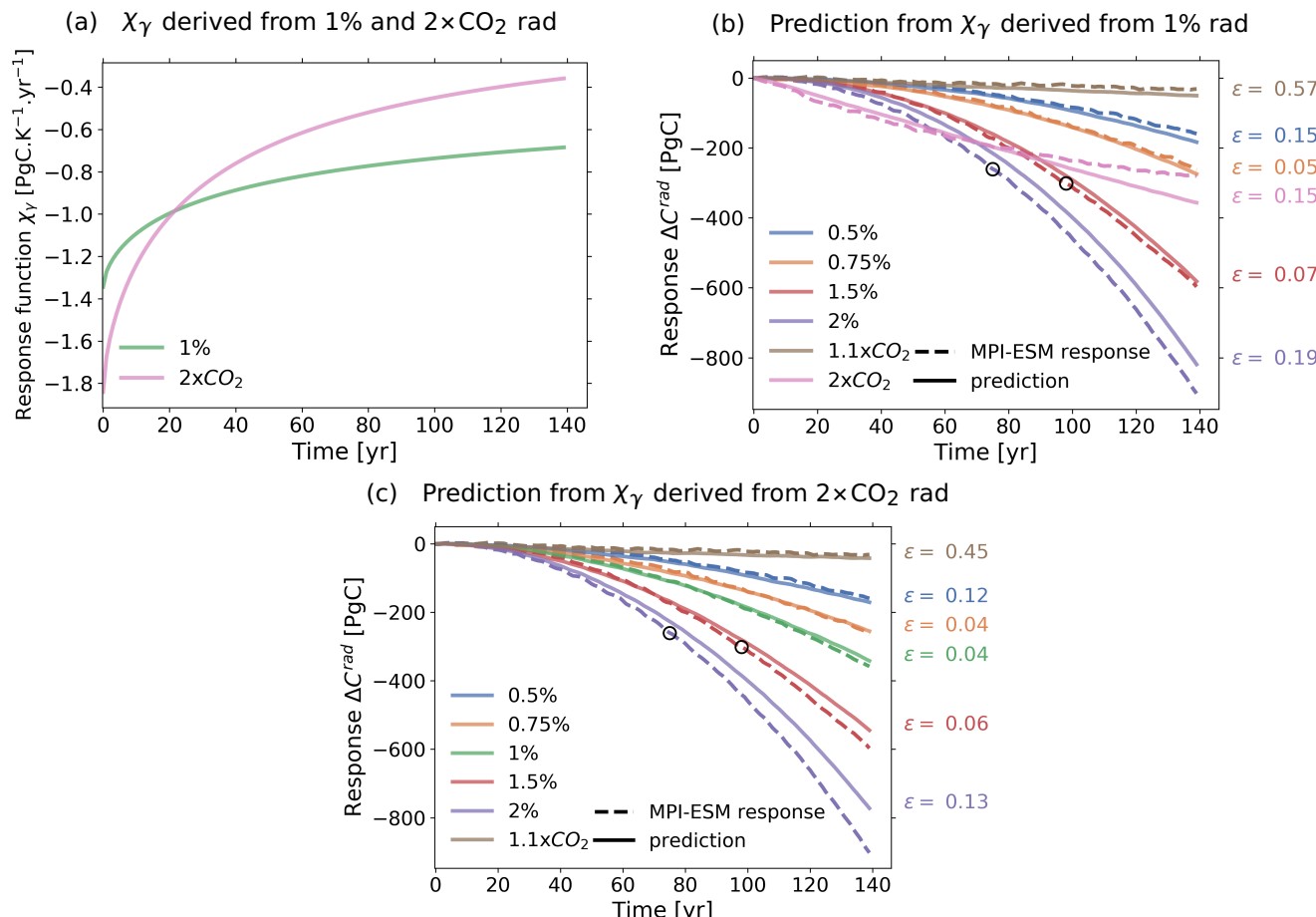

**Figure 5.** $\chi_\gamma$ recovered from 1% and 2×CO₂ rad experiments and prediction of model responses using these recoveries of $\chi_\gamma$. Circles indicate the maximum value for which 1.5% and 2% responses are predictable according to the estimate of the linear regime (see text). At the right of subfigures (b) and (c) the prediction error (see Eq. (7)) is indicated for the different experiments, calculated for the 1.5% and 2% rad experiments by considering only values preceding the circles.





predict a noisy response only with some finite error. Therefore, if $\chi_\gamma(t)$ is well recovered, we expect large prediction errors for experiments with small forcing strengths such as the $1.1\times CO_2$ rad – which is indeed the case –, but small errors for

experiments with large forcing strengths but still well within the linear regime such as the $2\times CO_2$ rad (compare the forcing strengths for the $2\times CO_2$ rad experiment and the maximum forcing strength for the 1% rad experiment in Fig. 4). Since contrarily to the expectation the prediction error is not particularly small for the $2\times CO_2$ rad experiment, probably the recovery of $\chi_\gamma(t)$ derived from the 1% rad experiment is not completely accurate and may still be further improved. As suggested above, such improvement may be achieved by taking data with better quality from the $2\times CO_2$ rad experiment.

Figure 5(c) shows the quality of the prediction using $\chi_\gamma(t)$ recovered from the $2\times CO_2$ rad experiment. As expected, results indicate an improvement in the recovery (compare to subfigure (b)). The prediction error decreases for all experiments present in both plots. In addition, it also decreases if we compare the prediction of the 1% rad response in subfigure (c) with that of the $2\times CO_2$ rad response in subfigure (b).

This section therefore suggests two main conclusions: First, for $\chi_\gamma(t)$ the response seems to be approximately linear for

temperature perturbations up to at least 6 K. Second, the overall improvement of the prediction in Fig. 5(c) compared to Fig. 5(b) confirms the expectation from the Laplace transform analysis that indeed the step-like $2\times CO_2$ rad experiment carries more information on the response function than the 1% rad experiment. This suggests that step-like experiments may be more appropriate than the standard 1% experiment for the identification of linear response functions.

## 4   Generalized sensitivity $\chi_\beta$

In this section we identify the linear response function $\chi_\beta$ (generalized $\beta$-sensitivity), defined by

$$\Delta C^{bgc}(t) = \int\limits_0^t \chi_\beta(t-s)\Delta c(s)ds + \eta(t), \tag{11}$$

where now the response is $\Delta Y(t) := \Delta C^{bgc}(t)$ and the forcing is $\Delta f(t) := \Delta c(t)$, with $\Delta C^{bgc}(t)$ being the change in global land carbon found in the "bgc" experiment and $\Delta c(t)$ the change in atmospheric $CO_2$.

That $\chi_\beta$ indeed generalizes $\beta$ can be understood analogously to section 3 by considering that $\beta$ is defined by

$$\beta(t) = \frac{\Delta C^{bgc}(t)}{\Delta c(t)} \overset{(11)}{=} \frac{1}{\Delta c(t)} \int\limits_0^t \chi_\beta(t-s)\Delta c(s)ds, \tag{12}$$

for a negligible noise $\eta(t)$. Hence, $\chi_\beta(t)$ can be seen as a generalization of $\beta(t)$ and a property of the land carbon cycle that characterizes the response $\Delta C^{bgc}(t)$ to any time-dependent perturbation $\Delta c(t)$, as long as the perturbation strength is small.





**Approaches to identify $\chi_\beta$**

Similarly to the last section, we identify $\chi_\beta(t)$ by several approaches to find the one that gives results with best quality. For

this purpose, we consider in addition to Eq. (11) also alternative formulas to derive $\chi_\beta(t)$, each taking a different forcing. The identification is performed in three different ways:

1. Using $CO_2$ as forcing (see Eq. (11));

2. Using the logarithm of $CO_2$ as forcing:

$$\Delta C^{bgc}(t) = \int_0^t \chi_\beta^{\ln}(t-s) c_{PI} \ln\left(\frac{c(s)}{c_{PI}}\right) ds + \eta(t),$$

(13)

where now the forcing is $\Delta f(t) := c_{PI} \ln\left(\frac{c(t)}{c_{PI}}\right)$, with $c_{PI}$ being the pre-industrial value for atmospheric $CO_2$;

3. Using Net Primary Production (NPP) as forcing:

$$\Delta C^{bgc}(t) = \int_0^t \chi_{NPP}(t-s) \Delta NPP(c(s)) ds + \eta(t).$$

(14)

The first approach is the same used throughout the paper: $\chi_\beta(t)$ is identified using the $\Delta c(t)$ forcing from Eq. (11).

For the second and third approaches, we take advantage of the fact that in the "bgc" setup, perturbations in atmospheric $CO_2$

affect land carbon only via changes in photosyntetic productivity. Therefore, we use the $CO_2$ forcing only indirectly via its relationship to NPP. By doing this, the hope is to account for some of the nonlinear contributions to the response that arise from this relationship. The advantage of accounting for these nonlinear contributions is that one can recover the response function from experiments with perturbation strengths larger than those possible when not accounting for nonlinear contributions (as in the first approach). Experiments with larger perturbation strengths give responses with higher SNR, which makes it possible to

recover the response function with better quality.

In the second approach (see Eq. (13)), we employ as forcing an explicit logarithmic expression describing the relationship between $CO_2$ and NPP (see e.g. Alexandrov et al., 2003). Such formula has the advantage that the expansion of the forcing in $c$ gives

$$\Delta C^{bgc}(t) = \int_0^t \chi_\beta^{\ln}(t-s) \Delta c(s) ds + \mathcal{O}((\Delta c)^2) + \eta(t).$$

(15)

Taking $\Delta c$ sufficiently small and comparing the result to Eq. (11) thus yields

$$\chi_\beta(t) = \chi_\beta^{\ln}(t).$$

(16)

Accordingly, the response function $\chi_\beta^{\ln}(t)$ from formula (13) gives as well the desired $\chi_\beta(t)$.





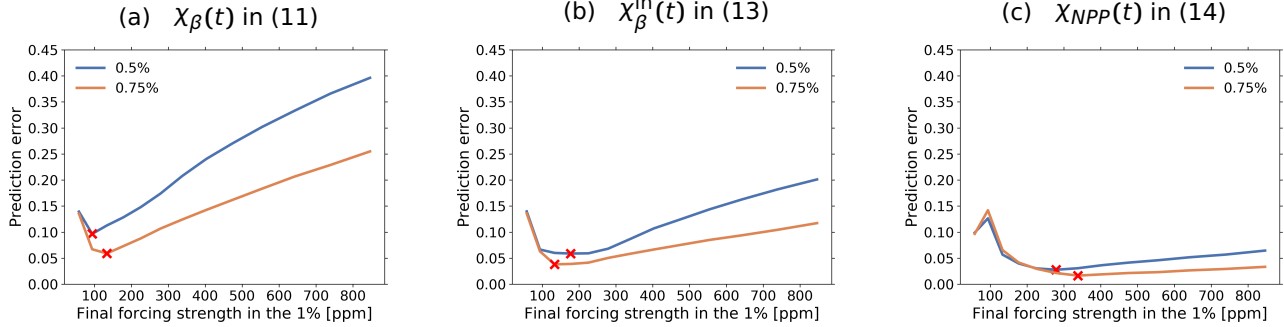

**Figure 6.** Prediction error (7) for the 0.5% and 0.75% bgc experiments obtained when using $\chi_\beta(t)$, $\chi_\beta^{\ln}(t)$ and $\chi_{NPP}(t)$ obtained from the 1% bgc experiment to predict the response. The error is shown as a function of the $CO_2$ final forcing strength.

In the third approach, we take directly the response to NPP (see Eq. (14)). Expanding the forcing $\Delta NPP(c)$ in $c$ gives

$$\Delta C^{bgc}(t) = \int_0^t \chi_{NPP}(t-s) \frac{\partial NPP}{\partial c}\Big|_{c=c_{PI}} \Delta c(s)ds + \mathcal{O}((\Delta c)^2) + \eta(t). \tag{17}$$

Taking $\Delta c$ sufficiently small and comparing the result to Eq. (11) yields

$$\chi_\beta(t) = \chi_{NPP}(t) \frac{\partial NPP}{\partial c}\Big|_{c=c_{PI}}. \tag{18}$$

Accordingly, by this approach we compute $\chi_\beta(t)$ in three steps: First, we identify the response function $\chi_{NPP}(t)$ using Eq. (14); second, we take the first derivative of NPP with respect to $CO_2$ at $c = c_{PI}$; third, we apply formula (18) to obtain $\chi_\beta(t)$ from $\chi_{NPP}(t)$.

**Checking nonlinearities with the three approaches**

Before analyzing the recovery of $\chi_\beta(t)$ employing the three approaches, one must verify that indeed these two additional approaches account for some of the nonlinearities in the response. If this is true, response formulas (13) and (14) should be able to predict the response to larger perturbation strengths than Eq. (11). To verify this expectation, in the following we compare the prediction error (7) by applying Eqs. (11), (13) and (14).

Figure 6(a) shows the prediction error for $\chi_\beta(t)$ recovered from the 1% bgc experiment with the first approach (Eq. (11)). The prediction is also computed via Eq. (11). The clear minima indicate the presence of strong nonlinearities for forcing strengths above around 100 ppm (94 ppm for the 0.5% and 133 ppm for the 0.75%). Therefore, in contrast to the case of $\chi_\gamma$ discussed in the last section, we see here that indeed one has to cope with the additional difficulty of nonlinearities.

Figure 6(b) shows the prediction error when using $\chi_\beta^{\ln}(t)$ recovered from the 1% bgc experiment with the second approach 325 (Eq. (13)). To check how well nonlinearities are accounted for by taking the logarithmic forcing, the prediction is as well computed via Eq. (13). Compared to subfigure (a), we see a slight improvement in the results: The minima have smaller





prediction errors and the prediction errors increase at a slower rate for increasing final forcing strength. This indicates that indeed using the logarithm of $CO_2$ as forcing accounts for some of the nonlinearities in the response. Accordingly, one can make predictions with smaller error for larger forcing strengths using Eq. (13) instead of Eq. (11).

Figure 6(c) shows the prediction error for $\chi_{NPP}(t)$ recovered via Eq. (14) from the 1% bgc experiment (first step of the third approach). To check how well nonlinearities are accounted for by taking the NPP forcing, we employ as well Eq. (14) for the prediction. Here, we see a substantial improvement in the results. The response is almost completely linear, with very "flat" minima. This indicates that indeed a large part of the nonlinearity encountered in the response of the land carbon to changes in $CO_2$ can be explained by the nonlinear relationship between NPP and $CO_2$. Accordingly, by employing Eq. (14) instead of

Eq. (11) one can predict the response of the land carbon until forcing strengths as high as 800 ppm with an error smaller than 10%.

### $\chi_\beta$ and the quality of its recovery

So far, we have only considered the ability of Eqs. (11), (13) and (14) to predict the land carbon response. Now, we analyze how well the generalized sensitivity $\chi_\beta(t)$ can be identified by the three approaches. For the identification we took data from

the 1% bgc experiment until the $CO_2$ forcing strength reaches the first minimum for each case in Fig. 6: $\Delta c$ = 94 ppm for the first approach (30 years); $\Delta c$ = 133 ppm for the second approach (40 years); and $\Delta c$ = 279 ppm for the third approach (70 years). Since $\Delta c$ = 279 ppm is approximately the forcing strength for the 2×$CO_2$ bgc experiment and results from last section suggested that this type of experiment may carry more information for the identification, we employ the third approach also taking the 2×$CO_2$ bgc experiment. For the present application where the recovery is limited by nonlinearities, taking

the 2×$CO_2$ bgc experiment has the additional advantage that because its forcing strength has a constant value throughout the whole experiment, we can use the full time series (140 years). To compute the first derivative of NPP with respect to $CO_2$ (second step of the third approach), we fitted the function $NPP = NPP(c)$ to polynomials of order 4, 5 and 6, and then took the first derivative from the fits.

      The results from the three approaches are shown in Fig. 7(a). At short time scales there is an overall agreement among all

recoveries with only small discrepancies. To be able to compare the results also for longer time scales, we extend the response functions recovered from the 1% bgc experiment – obtained from time series with 30, 40 and 70 years respectively for the first, second and third approaches – until 140 years (extensions are indicated by dotted lines). This can be done because with the RFI method we derive the response function from the ansatz (3), which formally gives the values of the response function for all times. The result is that all response functions recovered from the 1% bgc experiment present relatively small discrepancies

even at long time scales. Response functions derived from the 2×$CO_2$ bgc experiment with the third approach show a similar behaviour among themselves, but differ from the recoveries using the 1% bgc experiment. The reason for this difference will be investigated below.

      To quantitatively compare the quality of the recoveries, we plotted in Fig. 7(b) the prediction error (7). Since the response functions were derived using different time series lengths, for a fair comparison we compute the error only at the minimal

time series length of 30 years (the time series length used for the first approach). Results show no large discrepancies among


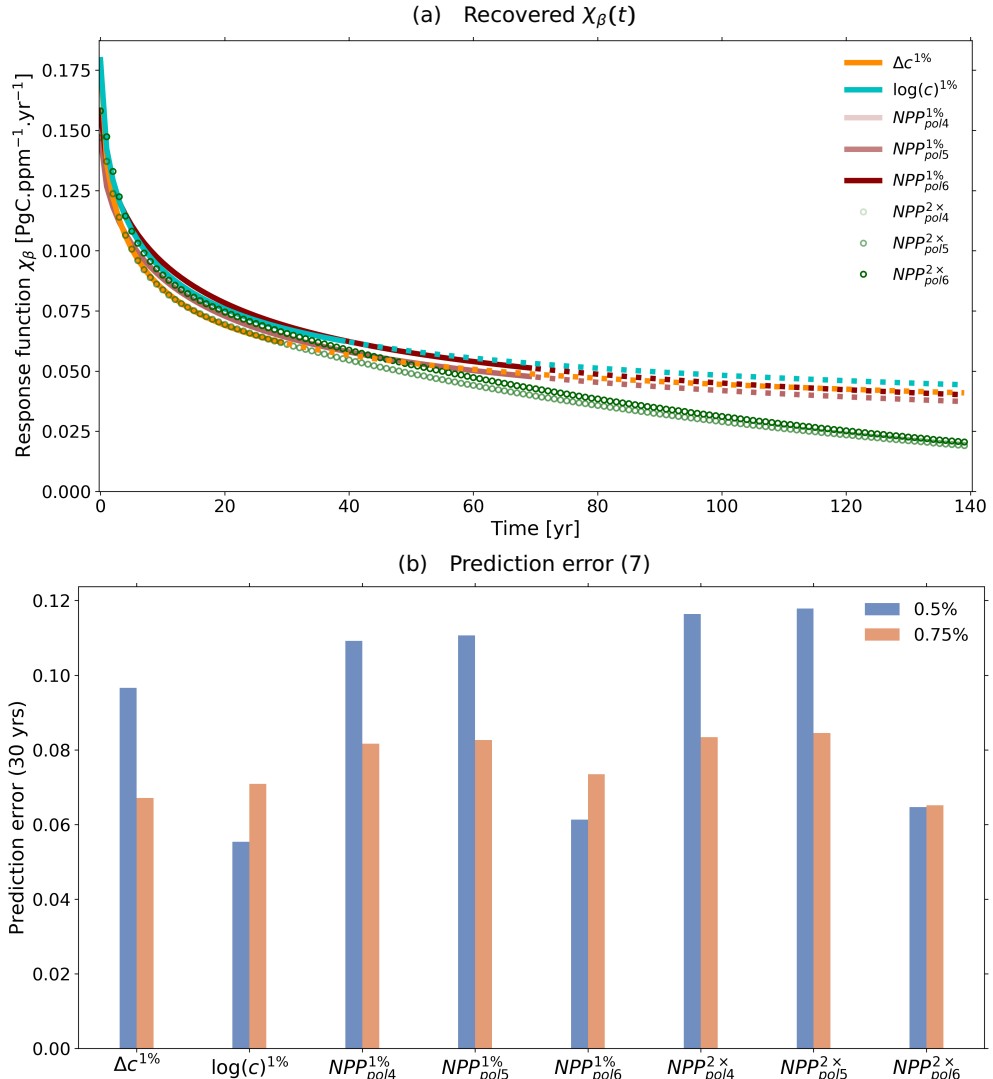

**Figure 7.** Response function $\chi_\beta(t)$ derived by the three approaches (subfigure (a)) and the respective prediction errors for the first 30 years of the response (subfigure (b)). $\Delta c^{1\%}$: recovery with first approach from 1% bgc experiment; $\log(c)^{1\%}$: recovery with second approach from the 1% bgc experiment; $NPP_{polx}^{1\%}$: recovery with third approach from the 1% bgc experiment using for the derivative a polynomial fit of order $x$; $NPP_{polx}^{2\times}$: recovery with third approach from the $2\times CO_2$ bgc experiment using for the derivative a polynomial fit of order $x$. Continuous lines denote values of $\chi(t)$ that are within the time series length used for the recovery (30 years for $\Delta c^{1\%}$, 40 years for $\log(c)^{1\%}$ and 70 years for $NPP_{polx}^{1\%}$). Dotted lines denote extended parts of the response function, i.e. values not covered by the time series used for the recovery but obtained from the recovered spectrum using Eq. (3). Circles denote the response functions derived taking the full time series ($NPP_{polx}^{2\times}$). For more details see text.





the different approaches. For the third approach, there seems to be a small advantage in using a polynomial of order 6 for the computation of the derivative.

But results from subfigure (b) reflect only the quality of the recoveries at short time scales, for which anyway no large discrepancies were encountered in subfigure (a). To evaluate the quality of the recovery also at long time scales, one must take

the extended version of the response functions that were derived from shorter time series. Since the only substantial difference at long time scales in Fig. 7(a) is found between the response functions recovered from the 1% and $2\times CO_2$ bgc experiments, we take for this analysis exemplarily only the response functions recovered with the first approach (1% bgc experiment) and the third approach ($2\times CO_2$ bgc experiment). By choosing the response function recovered with the first approach, we evaluate for the worst case scenario (where only 30 years are used for the recovery) how reliable predictions are for time scales longer than

the time series used for recovery. In contrast, by choosing the response function recovered with the third approach from the $2\times CO_2$ bgc experiment, we check whether the different values at long time scales are actually an improvement in the recovery. As mentioned above, such improvement is expected because this response function was recovered taking the full time series (140 years).

Following the same procedure as in the last section, in Fig. 8 we show the quality of the prediction for different experiments

using the aforementioned recoveries of $\chi_\beta(t)$. Subfigure (a) shows the results for the predictions calculated via Eq. (11) using $\chi_\beta(t)$ recovered with the first approach. We take as an estimate of the linear regime forcing strengths smaller than the forcing strength at the first minimum in Fig. 6(a). The response values corresponding to this forcing strength are marked in Fig. 8(a) with circles. It is seen that although $\chi_\beta(t)$ was recovered using a time series of only 30 years, it predicts the $1.1\times CO_2$ bgc experiment with only 3% error over 140 years. Other experiments are predicted within the linear regime with error smaller than

10% with exception of the 2% bgc experiment, for which the error is around 14%. Since the $2\times CO_2$ has a constant forcing strength outside the linear regime already from the beginning, its prediction fails as expected for the whole time series.

In subfigure (a) we could only evaluate the quality of the long time scales of $\chi_\beta(t)$ by the prediction of the $1.1\times CO_2$ bgc experiment, because this is the only experiment which has forcing strengths within the linear regime over the whole time series. To check the ability of $\chi_\beta(t)$ to predict also other experiments at long time scales, we account for some of the nonlinearity

in the response by taking NPP instead of $CO_2$ as forcing (see discussion of Fig. 6). Therefore, we perform the prediction by employing Eq. (14) instead of Eq. (11). Since for employing Eq. (14) one needs $\chi_{NPP}(t)$ and not $\chi_\beta(t)$, we take the $\chi_\beta(t)$ derived with the first approach and compute $\chi_{NPP}(t)$ from it via Eq. (18). Because the conversion from $\chi_\beta(t)$ to $\chi_{NPP}(t)$ is a simple scaling, the time scales structure is maintained. Hence, we can evaluate the quality of the recovered $\chi_\beta(t)$ from the results given by the obtained $\chi_{NPP}(t)$. The prediction results computed via Eq. (14) with the obtained $\chi_{NPP}(t)$ are shown in

subfigure (b). Because errors at the minima in Fig. 6(c) are not substantially smaller than those at the maximum final forcing strength, we take as an estimate of the linear regime all values smaller than the last value of NPP for the 1% bgc experiment (marked with circles in the responses). Once again it is seen that although $\chi_\beta(t)$ was recovered using a time series of 30 years, after conversion to $\chi_{NPP}(t)$ almost all experiments can be predicted with less than 10% error for the whole time series. The 1.5% and 2% bgc experiments are predicted with errors of 17% and 4% within the linear regime. Results from subfigures (a)

and (b) therefore suggest that although nonlinearities do restrict the recovery from Eq. (11), taking experiments with forcing

**Figure 8.** Prediction of model responses employing response functions derived with the first approach from the 1% bgc experiment (derived from data with 30 years length, but extended to 140 years) and with the third approach from the $2\times CO_2$ bgc experiment (derived from data with 140 years length). (a) Prediction by Eq. (11) taking the response function $\chi_\beta(t)$ derived from the 1% bgc experiment with the first approach. (b) Prediction by Eq. (14) taking the response function $\chi_\beta(t)$ derived from the 1% bgc experiment with the first approach and converted to $\chi_{NPP}(t)$ by Eq. (18). (c) Prediction by Eq. (14) taking the response function $\chi_{NPP}(t)$ derived in the first step of the third approach (see explanation after Eq. (18)) taking data from the $2\times CO_2$ bgc experiment. Continuous lines are predictions and dashed lines are responses from the MPI-ESM. Circles indicate the maximum value for which responses are predictable according to our estimate of the linear regime (see text). The values printed to the right of the plots are the prediction errors (see Eq. (7)) calculated for each experiment, considering when applicable only values preceding the circles.

strengths within the linear regime for the recovery leads to reliable prediction results even for times reasonably longer than the length of the time series from which the response function was recovered.





Finally, we investigate whether the different values seen in Fig. 7(a) for the $\chi_\beta(t)$ recovered from the 2×CO$_2$ bgc experiment indeed reflect a better quality of recovery. Following the same reasoning that led to subfigure (b), since in the third

approach $\chi_\beta(t)$ is obtained from a scaling of $\chi_{NPP}(t)$, we evaluate the quality of $\chi_\beta(t)$ from the results given by $\chi_{NPP}(t)$. Accordingly, in subfigure (c) we plot the prediction via Eq. (14) using the $\chi_{NPP}(t)$ recovered from the 2×CO$_2$ bgc experiment. As expected, the overall prediction indeed improves compared to subfigure (b). Individual prediction errors decrease for all experiments with exception of the 1.1×CO$_2$. Since the response functions used for subfigures (b) and (c) differ only at long time scales, this improvement suggests that indeed obtaining $\chi_\beta(t)$ from the 2×CO$_2$ bgc experiment gives a better recovery

over these time scales. Further, because all recoveries are similar at short time scales (see Fig. 7(a)), overall this recovery shows the best quality.

## 5   Spectrum of land carbon time scales

A response function obtained with high quality not only results in more accurate predictions, but may also provide valuable information about the internal dynamics of the system. For the case of $\chi_\beta(t)$, we find that the recovery with best quality gives

a relatively detailed description of the spectrum of land carbon time scales (see Eq. (3)). In this section, we investigate (i) the robustness of this result and (ii) the explanation for the structure of the obtained spectrum. The robustness of this detailed spectrum must be analyzed because, as explained in Part I, the problem to recover the spectrum of time scales from data is ill-posed so that in principle it is not clear to what extent the recovered spectrum can be trusted. To investigate this robustness, we check whether the main characteristics of the spectrum recovered by our RFI algorithm can as well be obtained by two

independent methods: a Gregory-plot approach (Gregory et al., 2004) and an approach that combines regional responses for the tropics and extra-tropics. Since the best recovery of $\chi_\beta(t)$ was obtained from the response to NPP for the 2×CO$_2$ bgc experiment, in our investigations we will study only this case.

**Obtaining the detailed spectrum**

But before we investigate the recovered spectrum, we demonstrate how such a detailed structure may arise from a better

recovery of the response function. In Fig. 9(a) we plot the spectrum $q(\tau)$ of the response function used for Fig. 8(b), i.e. $\chi_\beta(t)$ recovered with the first approach (see beginning of section 4) and converted to $\chi_{NPP}(t)$ via Eq. (18). Because the data used for the recovery have a relatively low quality – the response function was recovered from the 1% bgc experiment taken for only 30 years –, regularization filters out most of the SVD components of the spectrum in Eq. (4). Since the low-index SVD components that are not filtered out tend to be smooth (Hansen, 1989, 1990, see also Part I), the final result of this filtering is

the smooth spectrum seen in Fig. 9(a). Obviously, although this smooth spectrum is a sufficiently good approximation to make the predictions shown in Fig. 8(b), it is not very informative about the internal time scale structure. Instead, the spectrum of the response function $\chi_{NPP}(t)$ used for Fig. 8(c) has a more detailed structure (Fig. 9(b)). In this case, the higher quality of the data used for the identification (2×CO$_2$ bgc experiment taken for 140 years) results in less filtering by regularization, thereby





revealing more details of the underlying spectrum. The result is a spectrum with two peak time scales, at around 4 and 100
years[1].

**Checking the robustness of the spectrum via a Gregory-plot approach**

Some trust in this result may be gained by constructing a type of "Gregory plot" (Gregory et al., 2004) for the land carbon
(subfigure (c)) – which can be done because our $2\times CO_2$ bgc experiment is a step-like experiment. The idea here is to try to
identify from an independent method important time scales in the response so that they can be compared against the spectrum
in subfigure (b). For this analysis, we plot the global land carbon against its first time derivative (this is the net land-atmosphere
carbon flux). Using the $2\times CO_2$ bgc experiment where the forcing is constant, the first time derivative vanishes as the land
carbon approaches a new equilibrium value. The rate at which the derivative changes can be associated to a time scale $\tau_i$,
which should show up with a large weight value $q_i$ in the spectrum. Interestingly, the plot shows that the transient behaviour
towards equilibrium develops approximately at two different rates: a higher rate from the starting point until around 3520
PgC, and a lower rate from 3520 PgC onwards. To determine these rate values, we fitted a linear function $dC/dt = a + bC^{bgc}$
for each of these two ranges of the land carbon. Then, from each rate value $b$ we computed a time scale by $\tau = -1/b$. The
computed time scales taking one standard deviation into account are shown by the two ranges highlighted in subfigure (b). As
seen, also the Gregory plot reveals a time scale structure dominated by two time scales. While the shortest time-scale peak of
the spectrum partially overlaps with the corresponding time-scale range from the Gregory plot, the longest time-scale peak is
almost perfectly matched[2]. This result suggests that indeed the recovered spectrum reflects internal characteristics of the global
response of the land carbon cycle.

**Checking the robustness of the spectrum via regional response functions**

The robustness of this detailed spectrum can be further checked by a different method. In the following, we test this robustness
by checking the consistency between the time scales of global and regional carbon responses.
To study regional carbon responses we considered two regions: Tropics and extra-tropics. Tropics were defined as the
region between latitudes 30° south and 30° north, and extra-tropics as the remaining part of the globe. We then determined
separately the linear response functions that characterize the land carbon response to NPP in the tropics and extra-tropics,

---

[1]We ignore the negative values obtained for time scales smaller than 1 year (data time step) and larger than 140 years (time series length) because spectral
contributions at time scales much longer or much shorter than the time scales covered by data cannot be correctly recovered. Yet, as demonstrated in Appendix
D, their wrong recovery does not strongly affect the recovered response function so that they can be safely ignored. Other slightly negative values between the
two peaks are probably small recovery errors that inevitably appear as a consequence of ill-posedness and the filtering by regularization (such slightly negative
values have also been observed in recovered spectra shown in Appendix E).

[2]A possible reason for the better matching at long time scales is that these time scales contribute to the response at short and long times, while contributions
at short time scales decay rapidly and therefore contribute only at short times. This gets clear by considering for example a response function $\chi(t) =
a_1 e^{-t/\tau_1} + a_2 e^{-t/\tau_2}, \tau_2 \gg \tau_1$. While $a_2 e^{-t/\tau_2}$ contributes to $\chi(t)$ at small and long times, $a_1 e^{-t/\tau_1}$ contributes only at small times.


**Figure 9.** Spectra associated to $\chi_{NPP}$ derived with different resolutions and Gregory plot for land carbon. (a) Spectrum derived with the first approach (taking $CO_2$ as forcing) from the 1% bgc experiment; (b) Spectrum derived with the third approach (taking NPP as forcing) from the $2\times CO_2$ bgc experiment; (c) "Gregory plot" for land carbon. Dots are the data, with the color scale from white to black indicating the evolution from 1 to 140 years. Values of $b$ indicate the rate at which the time derivative of land carbon changes with respect to the land carbon itself. Ranges of time scales corresponding to each rate accounting for one standard deviation are shown in subfigure (b).

defined respectively by

$$\Delta C^{bgc,tr}(t) = \int_0^t \chi_{NPP}^{tr}(t-s)\Delta NPP^{tr}(s)ds + \eta^{tr}(t), \tag{19}$$

$$\Delta C^{bgc,et}(t) = \int_0^t \chi_{NPP}^{et}(t-s)\Delta NPP^{et}(s)ds + \eta^{et}(t). \tag{20}$$

**Figure 10.** Investigation of the land carbon response in the tropics and extra-tropics and how the regional response functions combine to the global response functions. The analysis is based on the $2\times CO_2$ bgc experiment. (a) Laplace transform $\widetilde{\chi}_{NPP}(p)$ of global $\chi_{NPP}(t)$ obtained directly from the global carbon response and from combining the tropical and extra-tropical response functions; (b) $\chi_\beta(t)$ obtained directly from the global carbon response and from combining the tropical and extra-tropical response functions; (c) As (b) but for $q_\beta(\tau)$; (d) Decomposition of $q_\beta(\tau)$ into tropical and extra-tropical spectra (Eq. (28)). In (c) and (d) the dots and crosses indicate the computed values, while the connecting lines are only inserted to guide the eye. For more details see text.

The data were taken once more from the $2\times CO_2$ bgc experiment.

Before assessing the robustness of the spectrum, we check the consistency between the regional and global response functions. In this test, we show that the global response function $\chi_{NPP}$ can be reconstructed by combining $\chi_{NPP}^{tr}$ and $\chi_{NPP}^{et}$. For





this purpose, we write the global land carbon as the sum of the land carbon in the tropics and extra-tropics:

$$\Delta C^{bgc}(t) = \Delta C^{bgc,tr}(t) + \Delta C^{bgc,et}(t). \tag{21}$$

Plugging Eqs. (14), (19) and (20) in Eq. (21) and recognizing that $\eta(t) = \eta^{tr}(t) + \eta^{et}(t)$ gives

$$\int_0^t \chi_{NPP}(t-s)\Delta NPP(s)ds = \int_0^t \chi_{NPP}^{tr}(t-s)\Delta NPP^{tr}(s)ds + \int_0^t \chi_{NPP}^{et}(t-s)\Delta NPP^{et}(s)ds. \tag{22}$$

Applying a Laplace transform to both sides of Eq. (22) and reorganizing the resulting equation gives

$$\widetilde{\chi}_{NPP}(p) = \frac{\widetilde{\chi^{tr}}_{NPP}(p)\widetilde{\Delta NPP^{tr}}(p) + \widetilde{\chi^{et}}_{NPP}(p)\widetilde{\Delta NPP^{et}}(p)}{\widetilde{\Delta NPP}(p)}. \tag{23}$$

Therefore, the Laplace transformed $\widetilde{\chi}_{NPP}(p)$ can be obtained from combining the NPP forcings and the response functions for the tropics and extra-tropics. Hence, if the response functions are correctly recovered by the RFI algorithm, Eq. (23) should hold at least approximately. In order to check this, we computed the Laplace transforms analytically by approximating the NPP forcings by step functions (since we take the $2\times CO_2$ bgc experiment), and using Eq. (3) for the response functions. Figure 10(a) shows the quality of the result. The small discrepancy between $\widetilde{\chi}_{NPP}(p)$ obtained from the global carbon response and from combining the regional responses can be at least partially explained by the approximation error made to represent the forcings by a step function.

We now check the robustness of the land carbon spectrum by combining $\chi_{NPP}^{tr}(t)$ and $\chi_{NPP}^{et}(t)$ to obtain $\chi_\beta(t)$ and thereby $q_\beta(\tau)$. To this end, we first obtain the tropical $\chi_\beta^{tr}(t)$ and extra-tropical $\chi_\beta^{et}(t)$ by applying Eq. (18):

$$\chi_\beta^{tr}(t) = \frac{\partial NPP^{tr}}{\partial c}\Big|_{c=c_{PI}} \chi_{NPP}^{tr}(t), \tag{24}$$

$$\chi_\beta^{et}(t) = \frac{\partial NPP^{et}}{\partial c}\Big|_{c=c_{PI}} \chi_{NPP}^{et}(t). \tag{25}$$

Using the obtained response functions, one can now write Eq. (11) for global, tropical and extra-tropical carbon. Plugging the result into Eq. (21) gives

$$\int_0^t \chi_\beta(t-s)\Delta c(s)ds = \int_0^t [\chi_\beta^{tr}(t-s) + \chi_\beta^{et}(t-s)]\Delta c(s)ds. \tag{26}$$

Hence, one can infer that

$$\chi_\beta(t) = \chi_\beta^{tr}(t) + \chi_\beta^{et}(t). \tag{27}$$

Therefore, the global response function $\chi_\beta(t)$ can be obtained from $\chi_\beta^{tr}(t)$ and $\chi_\beta^{et}(t)$. But in addition, since $\chi(t)$ is given by Eq. (3), Eq. (27) implies that one can also obtain the global spectrum $q_\beta(\tau)$ by combining the regional spectra:

$$q_\beta(\tau) = q_\beta^{tr}(\tau) + q_\beta^{et}(\tau). \tag{28}$$





Therefore, using the recovered response functions for tropical and extra-tropical carbon one can obtain the global response

function $\chi_\beta$ and its associated spectrum $q_\beta$. Accordingly, in this test we check Eqs. (27) and (28). For the calculation of the derivatives in Eqs. (24) and (25) we fitted $NPP^{tr} = NPP^{tr}(c)$ and $NPP^{et} = NPP^{et}(c)$ once again by a polynomial of order 6 (which obtained the best results for global NPP in Fig. 7(b)) and took the derivatives from the fits. For $\chi_\beta(t)$ we used the recovery with best quality from Fig. 7 ("$NPP^{2\times}_{pol6}$"). The spectra $q_\beta(\tau)$, $q_\beta^{tr}(\tau)$ and $q_\beta^{et}(\tau)$ are obtained by scaling the spectra from $\chi_{NPP}(t)$, $\chi_{NPP}^{tr}(t)$ and $\chi_{NPP}^{et}(t)$ by the respective derivatives $\frac{\partial NPP}{\partial c}\big|_{c=c_{PI}}$, $\frac{\partial NPP^{tr}}{\partial c}\big|_{c=c_{PI}}$ and $\frac{\partial NPP^{et}}{\partial c}\big|_{c=c_{PI}}$.

Results are shown for $\chi_\beta$ in Fig. 10(b) and for $q_\beta$ in Fig. 10(c). As seen in subfigure (b), the reconstruction matches almost perfectly the values of $\chi_\beta$ for times beyond about 20 years. Likewise, the spectrum $q_\beta$ is almost perfectly reconstructed for time scales above 6 years, with a slight discrepancy between 15 and 25 years (see Fig. 10(c)). A larger disagreement is seen below 20 years for $\chi_\beta$ and below 6 years for $q_\beta$. One of the reasons is that only little information is available for time scales smaller than 1 year because this is the time step taken for the data. However, Appendix D shows that time scales much shorter than the

time step affect only $\chi(0)$. The main reason for the disagreement is that high frequencies (and thus small time scales) are the most problematic to recover due to the ill-posedness of the problem that obscures the signal particularly at high frequencies.

Despite the disagreement at small time scales, Figs. 10(b) and (c) add confidence that: 1) The response functions for global, tropical and extra-tropical carbon can be trusted over mid-to-long time scales (Fig. 10(b)); and 2) the two-peak spectrum obtained for global land carbon indeed characterizes the response, since its computation from two independent approaches

(combining regional spectra in Fig. 10(c) and the Gregory plot in Fig. 9(c)) yield similar results with characteristic time scales matching the peaks of the spectrum.

**Investigating the two-peak structure of the spectrum**

The reasons for the two-peak structure of the land carbon spectrum are conceivable. In principle, one possibility could be that the short time scales reflect the carbon dynamics in the tropics, a region with higher temperatures and thus larger heterotrophic

respiration rates (see e.g. Raich and Potter, 1995), while the long time scales may originate from the carbon dynamics in the extra-tropics, where respiration rates are smaller due to lower temperatures. This hypothesis may be checked by examining Fig. 10(d), which shows the contribution from the tropics and the extra-tropics to the land carbon spectrum (Eq. (28)). But as seen, the two peaks arise both in the tropical and in the extra-tropical spectrum, so that one cannot attribute each peak to a particular region. Therefore, this cannot be the explanation.

An alternative hypothesis is that the different peak time scales originate from the very different characteristic time scales of functionally different elements in the land carbon cycle such as leaves, wood, litter and soil. In the following we investigate whether this hypothesis can explain the two-peak structure.

For this purpose, we split the land carbon pools of the MPI-ESM into two groups (see Fig. 11). In the first group are the pools whose dynamics is governed by fast processes such as shedding and decomposition of leaves, thus the pools that respond

at short time scales. These are the pools representing non-woody tissues in living vegetation (leaves, fine roots, sugars, and starches) and the associated litter. In the second group are the pools with dynamics determined by slow processes such as the



decomposition of woody plant parts, hence the pools that respond at long time scales. These are the pools representing the wood in stems and coarse roots, the associated litter pools, and the soil carbon (humus).

Now, we separate the land carbon spectrum into contributions arising from each of these two groups. The land carbon response can be described as the sum of the collective responses from the pools with fast and those with slow dynamics:

$$\Delta C^{bgc}(t) = \Delta C^{bgc,f}(t) + \Delta C^{bgc,s}(t). \tag{29}$$

In the linear response framework, the response of each term to NPP is given by

$$\int_0^t \chi_{NPP}(t-s)\Delta NPP(s)ds = \int_0^t \chi_{NPP}^f(t-s)\Delta NPP(s)ds + \int_0^t \chi_{NPP}^s(t-s)\Delta NPP(s)ds, \tag{30}$$

which implies

$$\chi_{NPP}(t) = \chi_{NPP}^f(t) + \chi_{NPP}^s(t). \tag{31}$$

Finally, assuming that each response function in Eq. (31) is described by Eq. (3), we obtain the separation of the land carbon spectrum in terms of the contribution from each pool group:

$$q_{NPP}(\tau) = q_{NPP}^f(\tau) + q_{NPP}^s(\tau). \tag{32}$$

Hence, if our hypothesis is correct, then the peak at short time scales should be explained by $q_{NPP}^f$ and the peak at long time scales should be explained by $q_{NPP}^s$.

To proceed with the analysis we now need to obtain the spectra $q_{NPP}^f$ and $q_{NPP}^s$. When investigating the tropical and extra-tropical land carbon, we obtained each regional spectrum individually by applying the RFI algorithm to the data from the tropical and extra-tropical responses. In principle one could proceed in the same way to separate contributions from the two pool groups, but, as will get clear below, this approach introduces slight inconsistencies between the separate recoveries that makes a quantitative comparison of the two contributions less reliable than the alternative method that we use in the following to separate the fast and slow components of $q_{NPP}$.

The idea of this alternative approach is the following. Numerically the land carbon spectrum is given by the regularized solution (4), i.e.

$$\boldsymbol{q}_{NPP,\lambda} = \sum_{i=0}^{M-1} \frac{\sigma_i^2}{\sigma_i^2 + \lambda} \frac{\boldsymbol{u}_i \bullet \Delta \boldsymbol{C}^{bgc}}{\sigma_i} \boldsymbol{v}_i, \tag{33}$$

where the regularization parameter $\lambda$ is determined by the RFI algorithm. The land carbon response $\Delta \boldsymbol{C}^{bgc}$ is given by the sum (29) of the responses from the pools with fast and those with slow dynamics. Entering Eq. (29) into Eq. (33) yields

$$\boldsymbol{q}_{NPP,\lambda} = \sum_{i=0}^{M-1} \frac{\sigma_i^2}{\sigma_i^2 + \lambda} \frac{\boldsymbol{u}_i \bullet (\Delta \boldsymbol{C}^{bgc,f} + \Delta \boldsymbol{C}^{bgc,s})}{\sigma_i} \boldsymbol{v}_i = \boldsymbol{q}_{NPP,\lambda}^f + \boldsymbol{q}_{NPP,\lambda}^s. \tag{34}$$





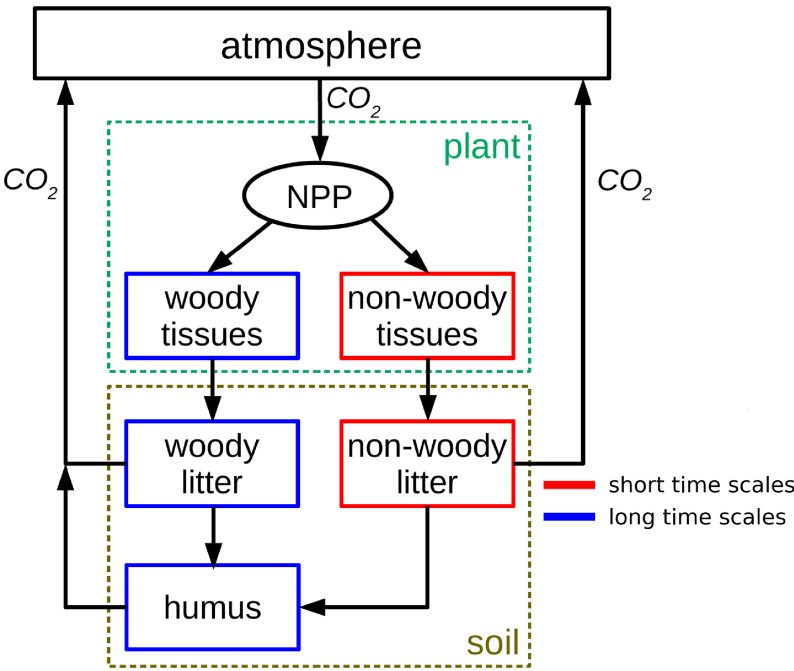

**Figure 11.** Simplified pool scheme of the land carbon cycle in the MPI-ESM (see Appendix A) with pools split into two groups according to the characteristic time scales underlying their dynamics.

Therefore, by deriving the spectra $\boldsymbol{q}^f_{NPP,\lambda}$ and $\boldsymbol{q}^s_{NPP,\lambda}$ using the same regularization parameter $\lambda$ employed for the land carbon spectrum $\boldsymbol{q}_{NPP,\lambda}$, we can in principle accurately reconstruct $\boldsymbol{q}_{NPP,\lambda}$ from $\boldsymbol{q}^f_{NPP,\lambda}$ and $\boldsymbol{q}^s_{NPP,\lambda}$.

By obtaining $\boldsymbol{q}^f_{NPP,\lambda}$ and $\boldsymbol{q}^s_{NPP,\lambda}$ in this way and combining them via Eq. (32) we show that indeed $\boldsymbol{q}_{NPP,\lambda}$ can be very accurately reconstructed (Fig. 12(a)). This approach leads as well to an almost perfect reconstruction of the response function $\chi_{NPP}$ via Eq. (31), as shown in Fig. 12(b). Compared to our previous approach employed to reconstruct the global spectrum from the regional spectra (Eq. (28)), this alternative approach gives a more accurate reconstruction because it takes the same regularization parameter $\lambda$ for all $\boldsymbol{q}_{NPP,\lambda}$, $\boldsymbol{q}^f_{NPP,\lambda}$, and $\boldsymbol{q}^s_{NPP,\lambda}$, in contrast to the previous approach where $\lambda$ was separately
calculated by the RFI algorithm for each spectrum in Eq. (28).

Now, the question is whether the spectra $q^f_{NPP}$ and $q^s_{NPP}$ can indeed explain each peak in the land carbon spectrum $q_{NPP}$. To check this, in Fig. 12(c) we plot the three spectra. As seen, clearly the peak at short time scales of the land carbon spectrum arises mostly from the large peak in the fast-dynamics spectrum, while the peak at long time scales follows closely the large peak in the slow-dynamics spectrum. This result thus indicates that our hypothesis is correct, so that indeed the two-peak
spectrum originates from the different characteristic time scales of functionally different elements in the land carbon cycle.



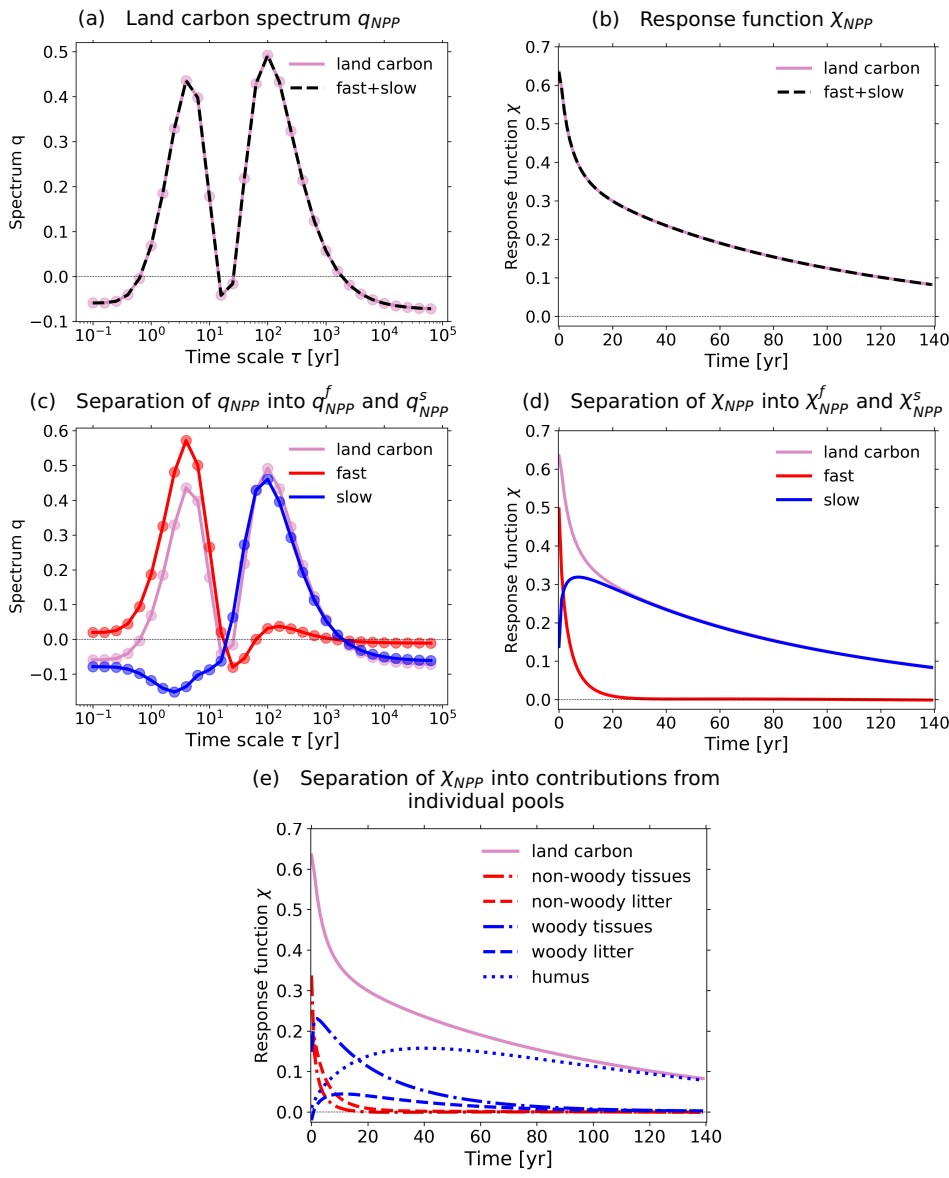

**Figure 12.** Investigation of the contribution from the pools with fast dynamics (non-woody pools) and those with slow dynamics (woody and humus pools) to the land carbon spectrum. The analysis is based on the $2 \times CO_2$ bgc experiment. (a) Land carbon spectrum $q_{NPP}$ and its reconstruction via Eq. (32) from the fast and slow components; (b) response function $\chi_{NPP}$ and its reconstruction via Eq. (31) from the response functions for the fast and the slow components; (c) separation of spectrum $q_{NPP}$ into $q_{NPP}^f$ and $q_{NPP}^s$; (d) separation of response function $\chi_{NPP}$ into $\chi_{NPP}^f$ and $\chi_{NPP}^s$; (e) separation of $\chi_{NPP}$ into contributions from the individual pools. For more details see text.





More insight into the dynamics of the land carbon can be gained by analyzing the different contributions to the response function $\chi_{NPP}$ shown in Figs. 12(d) and 12(e). Subfigure (d) shows the contribution of the pools with fast dynamics $\chi_{NPP}^{f}$ and that of the pools with slow dynamics $\chi_{NPP}^{s}$. We see that $\chi_{NPP}^{f}$ only dominates the response at times smaller than about 2 years. Further, for times larger than 25 years only $\chi_{NPP}^{s}$ contributes to the response. In subfigure (e) we further separate the

response functions into the contributions from the individual pools, i.e.

$$\chi_{NPP}^{f}(t) = \chi_{NPP}^{nwt}(t) + \chi_{NPP}^{nwl}(t), \tag{35}$$

$$\chi_{NPP}^{s}(t) = \chi_{NPP}^{wt}(t) + \chi_{NPP}^{wl}(t) + \chi_{NPP}^{h}(t), \tag{36}$$

where $\chi_{NPP}^{nwt}$ is the response function for the non-woody tissues in living vegetation, $\chi_{NPP}^{nwl}$ for non-woody litter, $\chi_{NPP}^{wt}$ for woody tissues in living vegetation, $\chi_{NPP}^{wl}$ for woody litter, and $\chi_{NPP}^{h}$ for the humus pool. This more detailed separation

shows that for long times the humus pool starts to dominate the response so that at times larger than 100 years it gives the only contribution.

## 6   Summary, discussion and outlook

Although the $\gamma$ and $\beta$ values introduced by Friedlingstein et al. (2003) provide a useful framework for model intercomparison, they only characterize the sensitivity of a model for a particular perturbation scenario. Instead, one would like to characterize the

sensitivity as such. The dependence of $\gamma$ and $\beta$ on the considered scenario arises because of the internal memory of the system, i.e. because of the dependence of the response on past values of the perturbation. When the memory is taken into account, linear response functions arise as natural generalization of the $\gamma$ and $\beta$ sensitivities. But a fundamental step for applying this generalized framework is to identify the appropriate linear response functions. In Part I, we developed a method to identify linear response functions from data using only information from an arbitrary perturbation experiment and a control simulation.

In that study, the robustness of the method in the presence of noise and nonlinearity was demonstrated in applications to data from perturbation experiments performed with a toy model.

**Generalized land carbon sensitivities in the MPI-ESM**

In the present study, we demonstrated that our RFI method can as well be employed to derive response functions from C$^4$MIP data. Here, we identified for the MPI-ESM, using data from standard 1% experiments, the land carbon generalized sensitivities

$\chi_{\gamma}$ and $\chi_{\beta}$, i.e. the linear response functions that generalize the $\gamma$- and $\beta$-sensitivities for land carbon. The robustness of the identified generalized sensitivities was demonstrated by their ability to predict the response from experiments not used for the identification (sections 3 and 4). With the aid of additional experiments, we estimated the linear regime that gives the range of forcing strengths for which each generalized sensitivity can predict model responses. For $\chi_{\gamma}$, results indicate a linear response at least until the end of the 1% experiment, corresponding to temperature perturbations of around 6 K. For $\chi_{\beta}$, the estimate is

for CO$_2$ perturbation strengths up to 100 ppm. In addition, we analyzed different approaches that may be employed to improve the quality of the recovery of the response functions. For $\chi_{\gamma}$, taking the response from a 2×CO$_2$ experiment demonstrated to





give smaller prediction errors for every response evaluated, suggesting that this type of experiment gives also a better recovery. For $\chi_\beta$, we used the knowledge of the relationship between NPP and $CO_2$ to account for some of the nonlinearities in the response. We found that nonlinearities can to a large extent be explained by the nonlinear relationship between NPP and $CO_2$:

By using NPP instead of $CO_2$ as forcing an approximately linear response is found over the whole 1% experiment, i.e. until $CO_2$ perturbations of about 850 ppm. In addition, this approach yielded the best recovery for $\chi_\beta$. Evidences for this conclusion are the quality of its prediction and the detailed spectrum it yields for the response.

**Spectrum of land carbon time scales**

Obtaining the spectrum of the response is an additional advantage of the RFI method. Most methods in the literature either

recover $\chi(t)$ pointwise – and therefore do not give the spectrum –, or try to fit it with few exponents without accounting for ill-posedness, which does not give reliable results for the spectrum (see e.g. the famous example from Lanczos, 1956, p. 272). In the application to the MPI-ESM, obtaining the spectrum has proven advantageous for two reasons. First, it allows to formally extend the recovery of the response function beyond the time range of the length of the underlying time series. Results from such an extension (Figs. 8(a) and (b)) demonstrated that the recovered response function contains information on

times reasonably longer than the time series length used for the recovery.

Second, the spectrum gives valuable insight into the internal dynamics of the system. In particular, for our application the spectrum gives the most relevant time scales for the land carbon response on a global or regional level. The spectrum associated with the best recovery of $\chi_\beta$ showed two peak time scales for the global response: One around 4 and another around 100 years (section 5).

To obtain evidence for the robustness of this result, we showed that it is possible to reconstruct the global spectrum from the recovered tropical and extra-tropical spectra (see Fig. 10(c)), and that similar time scale ranges can be obtained via a "Gregory plot" approach (see Fig. 9(b) and (c)). Further, we demonstrated that the recovered tropical and extra-tropical response functions combine to the identified global response functions, indicating consistency between regional and global recovery (see Fig. 10(a) and (b)).

We then proceeded to investigate the reason for the two-peak structure of the spectrum. To this end, we separated the land carbon response into the response from pools with fast dynamics (non-woody vegetation tissues and associated litter) and the response from pools with slow dynamics (woody vegetation tissues, woody litter, and humus). By analyzing the spectrum for each of these responses we showed that the peak at short time scales of the land carbon spectrum arises mostly from the contribution of the pools with fast dynamics, while the peak at long time scales follows closely the contribution from the pools

with slow dynamics. This investigation therefore suggests that the two-peak spectrum results from the different contributions of functionally different elements of the land carbon cycle. Analysis of the response functions showed that the pools with fast dynamics dominate the land carbon response only for times below 2 years. Further, for times larger than 25 years only the pools with slow dynamics contribute to the response, and from 100 years onwards the contribution comes solely from the humus pool.





We remark that results for the spectrum should be always interpreted with care, since deriving the spectrum may be a more complicated problem than deriving the response function. This is partially explained by the degree of ill-posedness of each problem. In deriving the response function from Eq. (1), one is solving a deconvolution problem, which is known to be ill-posed because of the smoothing property of the convolution operator (e.g., Landl et al., 1991; Bertero et al., 1995; Hansen, 2002). On the other hand, by deriving the spectrum from Eqs. (1) and (3), one is solving not only a deconvolution (Eq. (1))

but also a type of inverse Laplace transform (to obtain the spectrum from the response function (3)), which is known to be extremely ill-posed because of the smoothing property of the Laplace operator (e.g., McWhirter and Pike, 1978; Istratov and Vyvenko, 1999). As a result, deriving $q(\tau)$ from the data $\Delta Y(t)$ involves two smoothing operations, namely the Laplace transform and convolution, whereas deriving $\chi(t)$ from $\Delta Y(t)$ involves only convolution, so that the problem of finding $q(\tau)$ may be more ill-posed than that of finding $\chi(t)$. This difficulty was exemplified by results from appendices D and E, which

discuss cases where the response function can be perfectly recovered but the recovered spectrum is relatively poor.

Another worthwhile comment is that, in comparison to the widely used assumption that the spectrum can be described by only a few exponents (Maier-Reimer and Hasselmann, 1987; Enting and Mansbridge, 1987; Enting, 1990; Joos et al., 1996; Pongratz et al., 2011; Joos et al., 2013; Colbourn et al., 2015; Lord et al., 2016), our assumption of a continuous spectrum (ansatz (3)) has some advantages. First, by our approach one circumvents the complication of determining the number of

exponents. This leads to a linear problem and thereby to a more transparent method (see Part I, section 3.1). Second, our ansatz better describes the expectation that variables integrated over many different climate zones are composed by a large range of time scales. If, in contrast to these expectations, the underlying spectrum has only few time scales, results in Appendix E show that our approach may still yield a reasonable recovery.

**Outlook**

In this study we investigated the generalized $\gamma$- and $\beta$-sensitivities that solve the scenario dependence problem noted e.g. by Gregory et al. (2009) and Arora et al. (2013) to linear order in the perturbation – an approach that can in principle be extended to higher orders (Ruelle, 1998; Lucarini, 2009). By demonstrating how to successfully recover generalized sensitivities, this study paves the way for their future application in studying the dynamics of the combined carbon-climate system in different Earth System Models. Applying our RFI method, we showed for the MPI-ESM that one can obtain these sensitivities from standard

C[4]MIP 1% simulations. In addition, the estimates for the linear regime obtained by employing the recovered sensitivities to predict additional experiments give a hint into the range of perturbation strengths for which this generalized framework might be valid as well for other models, for which data from additional experiments necessary to determine this linear regime are not available. Since the process descriptions of the land carbon cycle are quite similar in different models, it may be assumed that the linear range estimates obtained for MPI-ESM in the present study apply also to other models. Considering the radiative

land carbon response, thereby also for other models their $\chi_\gamma$ should fully characterize the response to temperature up to at least 6K. As a consequence, in models with temperature response similar or weaker than the MPI-ESM, $\chi_\gamma$ can as for the MPI-ESM be recovered taking the full time series from the 1% experiment. Analogously, considering the generalized sensitivity $\chi_\beta$, the experience presented in the present study for the MPI-ESM suggests that also for other models the response to $CO_2$




perturbations is linear up to 100 ppm. As a consequence, $\chi_\beta$ can be recovered taking data from the 1% experiment for the

first 30 years. Alternatively, as discussed in section 4, the time range for the recovery of $\chi_\beta$ can be extended if one takes as forcing NPP instead of $CO_2$ (third approach investigated in section 4). For models with NPP response similar or weaker than the MPI-ESM, by this approach $\chi_\beta$ can be recovered taking the full time series from the 1% experiment. Still, as the relatively successful predictions in Fig. 8(b) suggest, even if $\chi_\beta$ is recovered by taking $CO_2$ as forcing for only 30 years of the 1% experiment (first approach investigated in section 4), it may still contain sufficient information to predict responses of the

model for a time range of 140 years.

But in addition to the carbon-cycle sensitivities considered here, our RFI method may be applied to investigate different aspects of climate and the carbon cycle. Ragone et al. (2016) have shown how the linear response framework generalizes the concept of equilibrium climate sensitivity (ECS) and transient climate response (TCR). Originally, ECS is defined as the response of global temperature to an instantaneous doubling of atmospheric $CO_2$ from its pre-industrial value, while TCR is

the temperature response to a doubling of $CO_2$ after a 1% per year increase. Within the generalized framework, the ECS and TCR are shown to be only particular values encoded in a linear response function.

Linear response functions can also be useful in studying "committed changes" (Wigley, 2005; Plattner et al., 2008; Jones et al., 2009; Mauritsen and Pincus, 2017). As shown by Ragone et al. (2016), the concept of climate inertia (closely related to commited changes) can be explicitly described within the linear response framework. Since the linear response function

describes the delayed response of the system to a perturbation, by deriving this function one has at hand also the information of how the system reacts once the perturbation – for instance $CO_2$ emissions – ceases.

Further, linear response functions can help understanding the concept of "emergent constraints" (Nijsse and Dijkstra, 2018). Recent studies have shown how to obtain from response functions derived from conceptual models emergent constraints for the real Earth System (Cox et al., 2018; Williamson et al., 2019). With the method developed here, in principle such type of

analyses may be carried out employing instead response functions derived from Earth System Models.

As a result of accounting for the memory of the system, the linear response function gives information on the strength of the response at all internal time scales covered by the time series underlying its recovery and probably even slightly longer (see Fig. 8(b)). Using our method, in principle one can even compare the spectra of time scales from models to those from observations (e.g., Forney and Rothman, 2012b). The method presented here may also be applied to analyze changes in age and

transit time distributions of carbon in different models, since these distributions can be derived directly from linear response functions (Thompson and Randerson, 1999).

In all of the mentioned examples, the generality of the RFI method allows for the derivation of the appropriate linear response functions for any model by taking only data from an arbitrary perturbation experiment and a control experiment. Such generality opens the possibility of combining the linear response framework, which has been gaining increasing attention

due its wide applicability in climate science (e.g., Lucarini, 2009; Lucarini and Sarno, 2011; Lucarini et al., 2014; Ragone et al., 2016; Lucarini et al., 2017; Aengenheyster et al., 2018; Ghil and Lucarini, 2020; Lembo et al., 2020; Bódai et al., 2020), with model intercomparison studies, hopefully leading to a deeper understanding of critical differences encountered across models.





**Appendix A: The Max Planck Institute for Meteorology Earth System Model**

In this appendix we give a brief description of the model employed in this study: the Max Planck Institute for Meteorology Earth System Model (MPI-ESM; for more details see Giorgetta et al., 2013). The MPI-ESM consists of the coupled general circulation models ECHAM6 (Stevens et al., 2013) for atmosphere, at T63/1.9° horizontal resolution with 47 vertical levels, and MPIOM (Jungclaus et al., 2013) for ocean, with a nominal resolution of 1.5° with 40 vertical levels. In addition, the MPI-ESM includes the subsystems JSBACH (Reick et al., 2013; Schneck et al., 2013), a land and vegetation model, and the marine

biogeochemistry model HAMOCC (Ilyina et al., 2013). JSBACH and HAMOCC describe respectively the land and ocean carbon cycle in the MPI-ESM. Thus, of particular interest for our study is the subsystem JSBACH. JSBACH simulates fluxes of energy, water, momentum and $CO_2$ between the land surface and atmosphere. To represent subgrid scale heterogeneity, each grid cell of the land surface is divided into tiles, being each tile associated with a vegetation type (or "plant functional type"). The photosynthesis scheme follows Farquhar et al. (1980) and Collatz et al. (1992). The land carbon structure is divided into

three vegetation pools (living tissues, carbohydrate and starch storage, and wood), four aboveground and belowground pools for litter from woody and non-woody parts and a pool for soil carbon (humus). In addition, JSBACH includes a dynamic vegetation scheme that allows for simulating changes in vegetation cover driven by climate.

**Appendix B: Generalization of climate-carbon cycle sensitivities**

In this appendix we show that indeed linear response functions generalize Friedlingstein $\alpha$-$\beta$-$\gamma$ sensitivities. We explain this

by taking the $\beta$-sensitivity of land carbon uptake as an example. The calculation of $\beta$ is based on an experiment where the $CO_2$-rise acts only biogeochemically, i.e. concerning the land carbon via the photosynthesis of plants. Calling $\Delta C^{bgc}(t)$ the difference in land carbon between the perturbed and the control simulation, $\beta$ is defined as

$$\beta(t) = \frac{\Delta C^{bgc}(t)}{\Delta c(t)}, \quad \Delta c(t) = c_{PI}(1.01^t - 1),$$
(B1)

where $t$ is the time in years elapsed since the perturbation was switched on, and $\Delta c(t)$ is the change in $CO_2$ concentration

when increasing atmospheric $CO_2$ each year by 1% starting from its pre-industrial value $c_{PI}$ of the control simulation. In the framework of linear response, one can understand $\Delta c(t)$ as perturbation and $\Delta C^{bgc}(t)$ as response so that the response formula reads

$$\Delta C^{bgc}(t) = \int_0^t \chi_\beta(t-s)\Delta c(s)ds.$$
(B2)

This equation defines $\chi_\beta(t)$ as the linear response function describing the biogeochemical response of land carbon to any type

of $CO_2$ perturbation $\Delta c(t)$, as long as the perturbation is sufficiently weak. Employing in particular the percent perturbation from Eq. (B1), $\beta(t)$ is thus obtained from the linear response function $\chi_\beta(t)$ as

$$\beta(t) = \frac{1}{\Delta c(t)} \int_0^t \chi_\beta(t-s)\Delta c(s)ds.$$
(B3)





In this way, $\chi_\beta(t)$ can be understood as a generalization of $\beta(t)$, that gives not only the response to percent-type perturbations, but also to other perturbations. Accordingly, $\chi_\beta(t)$ characterizes system properties independent of the type of perturbation, in
contrast to $\beta(t)$.

**Appendix C: Monotonicity of $\boldsymbol{\chi_\beta(t)}$**

In this appendix it is shown that the response function $\chi_\beta(t)$ is monotonic, as claimed in subsection 2.1. The argument here is actually more general, namely that the response function to changes in the input of the system (for land carbon, the Net Primary Production) is monotonic. Since $\chi_\beta(t)$ is related to $\chi_{NPP}(t)$ by Eq. (18), the monotonicity property transfers to $\chi_\beta(t)$.
Let the linear response of land carbon be described by Eq. (14):

$$\Delta C^{bgc}(t) = \int_0^t \chi_{NPP}(t-s)\Delta NPP(s)ds, \tag{C1}$$

where for simplicity we assume that $\eta(t)$ is small so that it can be neglected. If NPP is a Dirac delta function $\Delta NPP(t) = \delta(t)$, then the response is given by $\Delta C^{bgc}(t) = \chi_{NPP}(t)$. Therefore we can interpret the response function $\chi_{NPP}(t)$ as follows: If a certain number of carbon atoms enter the biosphere at time $t = 0$, the response function $\chi_{NPP}(t)$ gives the fraction of these
atoms still left in the biosphere at time $t$.

Let $p(t)dt$ be the probability that an atom that entered the system at time $t = 0$ will leave it *at* time $t$. Then

$$P(t) := \int_0^t p(s)ds \tag{C2}$$

is the probability that an atom that entered the system at time $t = 0$ will leave the system *until* time $t$. Hence from the interpretation of $\chi_{NPP}(t)$ above
$$P(t) = 1 - \chi_{NPP}(t). \tag{C3}$$

From Eqs. (C2) and (C3) it follows that

$$p(t) = -\frac{d}{dt}\chi_{NPP}(t). \tag{C4}$$

But since the probability density function $p(t) \geq 0 \ \forall \ t$, then $\frac{d}{dt}\chi_{NPP}(t) \leq 0 \ \forall \ t$, i.e. $\chi_{NPP}(t)$ decays monotonically towards zero. Therefore, because $\chi_\beta(t)$ is simply a scaling of $\chi_{NPP}(t)$ given by Eq. (18), $\chi_\beta(t)$ also decays monotonically towards
zero.

**Appendix D: Derivation of spectrum and $\boldsymbol{\chi(t)}$ when the response contains time scales much longer or much shorter than the time scales covered by data**

Time scales much longer or much shorter than the time scales covered by data cannot be correctly recovered in the spectrum. Nevertheless, in this appendix we show that the wrong recovery of these extreme time scales does not strongly affect the





recovery of $\chi(t)$. These considerations add to the footnote remark in section 5 where we claim that such extreme time scales can be safely ignored.

First, we consider the case where the response has time scales much longer than those covered by data. Let $\chi$ at time $T$ be given by

$$\chi(T) = \int_{-\infty}^{\infty} q(\tau)e^{-T/\tau}d\log_{10}\tau. \tag{D1}$$

Let $T$ be the time series length and assume that $q(\tau)$ has significant contributions at time scales $\tau \geq \tau_L$ with $\tau_L >> T$. Then (D1) can be written as

$$\chi(T) \approx \int_{-\infty}^{\log_{10}\tau_L} q(\tau)e^{-T/\tau}d\log_{10}\tau + \int_{\log_{10}\tau_L}^{\infty} q(\tau)d\log_{10}\tau, \tag{D2}$$

where $e^{-T/\tau} \approx 1$ was used in the second integral because $\tau_L >> T$. Thereby the second term in the right-hand side of (D2) is just a constant offset

$$\int_{\log_{10}\tau_L}^{\infty} q(\tau)d\log_{10}\tau = k, \quad k \text{ constant.} \tag{D3}$$

Hence, for internal time scales much longer than the time series length $T$, the correct recovery of the individual $q(\tau)$ values is not relevant for the derivation of $\chi(t)$, as long as those values combine to the correct offset in Eq. (D3). Note that because this argument is based on the condition $\tau_L >> T$, it applies not only to $\chi(T)$ but also to all $\chi(t)$, $t < T$.

Now, we consider the case where the response has time scales much smaller than those covered by data. Let $t =: i\Delta t$, where 760   $\Delta t$ is the time step and $i \in IN$. If $q(\tau)$ has significant contributions at time scales $\tau \leq \tau_S$ with $\tau_S << \Delta t$, then for $i > 0$ Eq. (3) can be written as

$$\chi(i\Delta t) \approx \int_{\log_{10}\tau_S}^{\infty} q(\tau)e^{-i\Delta t/\tau}d\log_{10}\tau, \tag{D4}$$

where $e^{-i\Delta t/\tau} \approx 0$ was used for $\tau < \tau_S$ because $\tau_S << \Delta t$. As a result, values of $q(\tau)$ are irrelevant to $\chi(t)$ for almost every $i$. The only exception is $i = 0$, where one has

$$\chi(0) = \int_{-\infty}^{\infty} q(\tau)d\log_{10}\tau. \tag{D5}$$

Hence, for time scales much shorter than the time step $\Delta t$, the correct recovery of the individual $q(\tau)$ values is not relevant for the derivation of $\chi(t)$ as long as those values combine with the other recovered values of $q(\tau)$ to the correct $\chi(0)$ in Eq. (D5). As shown by Eq. (D4), if they combine to the wrong value, only the recovery of $\chi(0)$ is affected.

Therefore, in principle $\chi(t)$ can be correctly recovered also when the response contains much longer and much shorter 770   time scales than those covered by data, as long as the recovered $q(\tau)$ values at these extreme time scales combine to the





correct values of the offset in Eq. (D3) and $\chi(0)$ in Eq. (D5). In Fig. D1 we show an example of recovery of the spectrum and response function in such a case. For the recovery we took data with $SNR = 6 \times 10^4$ from the 1% experiment with our toy model (see Part I). As seen, the spectrum can only be partially recovered (subfigure (a)). The recovery wrongly estimates spectral contributions at time scales longer and shorter than those covered by data, i.e. time scales larger than the time series

length T = 140 and smaller than the time step $\Delta t = 1$. In addition, the existence of such spectral contributions at time scales not covered by data seems to also deteriorate the recovery of the spectrum at the time scales actually covered by data: In the region $1 < \tau < 140$, despite the high SNR the recovered spectrum matches the true spectrum only partially. This is probably a compensation effect where wrong information shows up in the recovered spectrum to compensate missing information on the response function. For instance, to obtain the correct $\chi(0)$, only Eq. (D5) is needed but not the correct recovery of all time

scales. The same goes for obtaining the correct offset: Only Eq. (D3) is needed but not the correct recovery of all time scales from $\tau_L$ onwards.

   Subfigure (b) shows the quality of the recovery of the response function $\chi(t)$. The recovered $q(\tau)$ values at long time scales, although individually wrong, combine to the correct offset in (D3): The "height" of the recovered response function matches almost perfectly that of the true response function (compare e.g. the value at t = 140 for the true and recovered response

function). On the other hand, $\chi(0)$ is not perfectly recovered (compare the difference at t = 0 between the true and recovered response functions), meaning that the sum in Eq. (D5) is incorrect. Nevertheless, the recovered value is still reasonably close to the true value. Except for this small error at $\chi(0)$, the overall recovery of the response function is almost perfect.

   Therefore, as claimed in section 5, this numerical example shows that even though very long and very short time scales cannot be correctly recovered in the spectrum, they do not strongly influence the recovery of $\chi(t)$. This is because they only

influence the offset in Eq. (D3) and $\chi(0)$ in Eq. (D5), and those two values seem to be reasonably well recovered numerically.

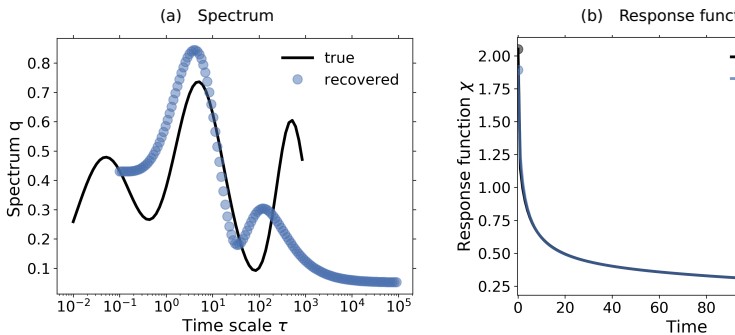

**Figure D1.** Recovery of spectrum and $\chi(t)$ taking a true spectrum with contributions at time scales much longer and much shorter than the time scales covered by data. For the recovery, data with $SNR = 6 \times 10^4$ from the 1% experiment with the toy model (see Part I) was taken (times series length T = 140 and time step $\Delta t = 1$). RFI parameters are taken as in Part I except for $M = 140$. The recovered spectrum matches only partially the true spectrum. Nevertheless, the response function $\chi(t)$ is almost perfectly recovered.





## Appendix E: Recovering a discrete spectrum $q(\tau)$

This appendix gives numerical evidence for the claim in section 6 that although the RFI method assumes that the response has a continuous spectrum of many time scales, for sufficiently good signal-to-noise ratio the recovered continuous spectrum may also give a reasonable approximation to an underlying discrete spectrum of only few time scales.

Before we give numerical examples, one must first understand the limitations of the recovery of the spectrum. As explained in Part I, the spectrum $q(\tau)$ can only be completely recovered (for sufficiently high SNR) if it is dominated by the first components of expansion (4). By Hansen's observation (Hansen, 1989, 1990, see Part I), this means that the spectrum must be dominated by low frequencies, i.e. it must be to some extent smooth. However, an underlying discrete spectrum implies a spectrum that is instead given by "spikes" in the time scale domain. Such "spikes" can only be described by high-frequency components of

Eq. (4). Therefore, a discrete spectrum can be only partially recovered. To obtain a sufficiently good recovery, the solution (4) must contain many components. Hence, the data must have a sufficiently high signal-to-noise ratio. The explanation for this conclusion is the following: If many components in Eq. (4) must be recovered, the regularization parameter must be small; but a small regularization parameter requires a small noise level (see Theorem 3.3.1 in Groetsch, 1984).

With this in mind, we show in Figs. E1–E9 that at least smooth approximations to an underlying discrete spectrum can be

obtained. For the results we took data with $SNR \sim \mathcal{O}(10^4) - \mathcal{O}(10^5)$ from different experiments performed with the toy model described in Part I. Since the aim is to recover discrete spectra, a larger number of time scales $M$ gives a better resolution. Therefore, we take $M = 140$. All other RFI parameters are taken according to Part I. Also, monotonicity needed not to be taken into account (step 6 of Fig. 1 in Part I).

Figures E1–E6 show results for taking data from a 1% experiment and Figs. E7–E9 from a $2 \times f_0$ experiment. We start

with one time scale $\tau = 37$ (Fig. E1). As expected, the spike cannot be perfectly recovered, but the recovery gives a smooth approximation to it, with peak coinciding approximately with the "true" value. On the other hand, the response function is almost perfectly recovered. This is a result of the ill-posedness of Eq. (3): In the same way that high frequencies of the spectrum are suppressed in the data by Eq. (1), they are also suppressed in the response function by Eq. (3) (see Groetsch, 1984, section 1.1). Therefore, both the true spectrum and its smooth recovery result in practically the same response function.

In Fig. E2, the true spectrum has two time scales, this time at $\tau = 7$ and $\tau = 100$. Similarly to Fig. E1, the time scales are recovered by a smooth approximation with peaks approximately at the true values. Also similarly to Fig. E1, the response function is almost perfectly recovered. A similar result is obtained if we take time scales that are a bit closer together, as seen in Fig. E3 ($\tau = 7$ and $\tau = 37$). Nevertheless, now the peak for the longer time scale is a worse approximation, and there is a slightly pronounced negative peak that does not reflect the true spectrum.

In Fig. E4, we take instead time scales $\tau = 37$ and $\tau = 100$. Here, the resolution is not sufficiently high for a recovery of each time scale separately. Instead, the recovered spectrum shows only one mode that spans both spikes. Once more the response function can be almost perfectly recovered. Taking the three time scales $\tau = 7$, $\tau = 37$ and $\tau = 100$ (Fig. E5), the smooth recovery shows only two modes: One at $\tau = 7$ with peak almost coinciding with the true value, and another with peak





in between $\tau = 37$ and $\tau = 100$. If one in addition considers a fourth time scale $\tau = 1$ (Fig. E6), once more the recovery shows
only two modes. But now the mode at shorter time scales displays a longer tail that spans both $\tau = 7$ and $\tau = 1$.

The situation changes when we take for the recovery the $2 \times CO_2$ experiment. According to the Laplace transform analysis
in section 3, data from this experiment should give more information on small time scales. Figure E7 shows that this is indeed
the case: In contrast to the recovery from Fig. E6, now the time scale $\tau = 1$ is also recovered. On the other hand, now there are
two small negative peaks that do not reflect information from the true spectrum. Also, the resolution is not sufficiently good to
recover separately the two peaks at long time scales. However, if the longest time scale is increased from $\tau = 100$ to $\tau = 228$
(Fig. E8), a small peak can be seen around this time scale. In addition, now the peak at $\tau = 37$ matches slightly better the
true value. In Fig. E9 we add another time scale, now at $\tau = 518$. We see that although this time scale cannot be recovered
separately from $\tau = 228$, the peak at long time scales is more pronounced. This is in contrast to the fact that the time series
used for the recovery reaches only $t = 140$, indicating that the method can recover information on time scales even longer than
the time series length. This is in agreement with the conclusions from section 4, where in one case $\chi_\beta(t)$ was recovered from
a time series with only 30 years but could recover responses for $t = 140$ years (see Fig. 8(b)).

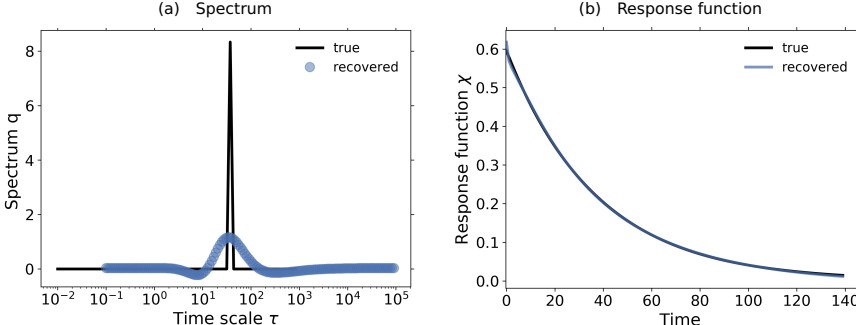

**Figure E1.** Spectrum $q_\lambda$ and response function $\chi(t)$ recovered from a 1% experiment performed with the toy model described in Part I
taking an underlying discrete spectrum with one time scale $\tau = 37$. The data were taken with $SNR \sim \mathcal{O}(10^4) - \mathcal{O}(10^5)$.





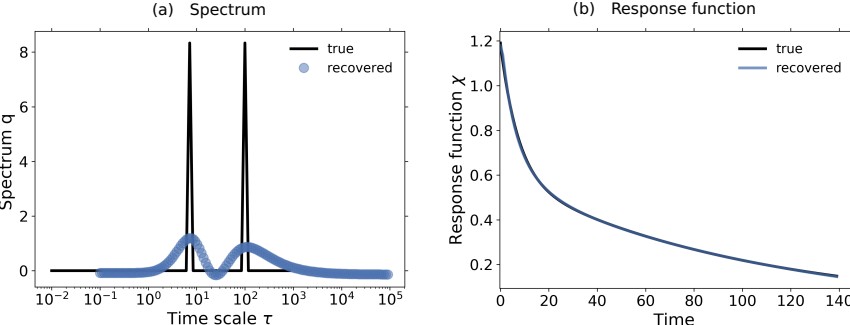

**Figure E2.** Spectrum $q_\lambda$ and response function $\chi(t)$ recovered from a 1% experiment performed with the toy model described in Part I taking an underlying discrete spectrum with two time scales $\tau = 7, \tau = 100$. The data were taken with $SNR \sim \mathcal{O}(10^4) - \mathcal{O}(10^5)$.

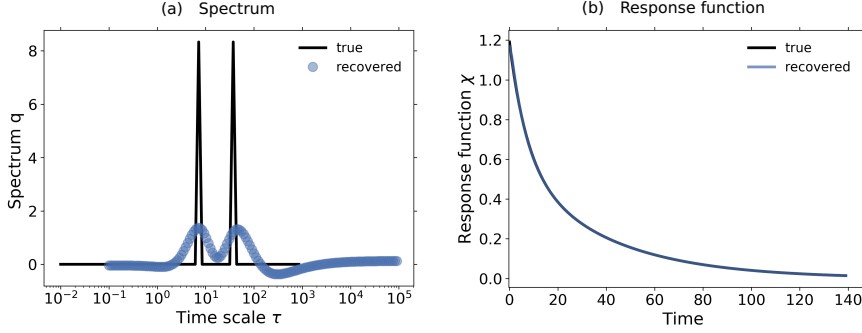

**Figure E3.** Spectrum $q_\lambda$ and response function $\chi(t)$ recovered from a 1% experiment performed with the toy model described in Part I taking an underlying discrete spectrum with two time scales $\tau = 7, \tau = 37$. The data were taken with $SNR \sim \mathcal{O}(10^4) - \mathcal{O}(10^5)$.

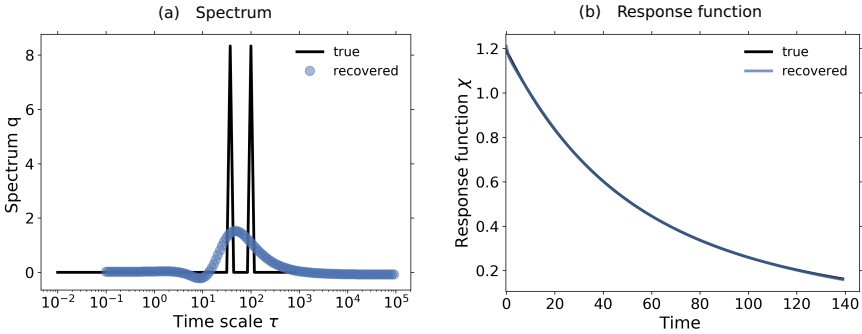

**Figure E4.** Spectrum $q_\lambda$ and response function $\chi(t)$ recovered from a 1% experiment performed with the toy model described in Part I taking an underlying discrete spectrum with two time scales $\tau = 37, \tau = 100$. The data were taken with $SNR \sim \mathcal{O}(10^4) - \mathcal{O}(10^5)$.



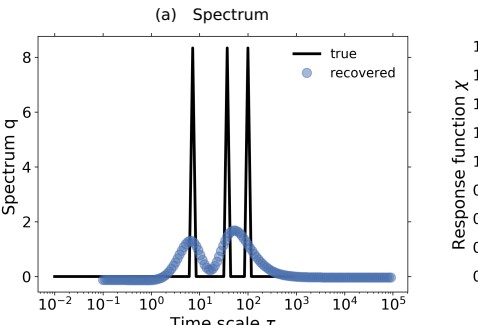
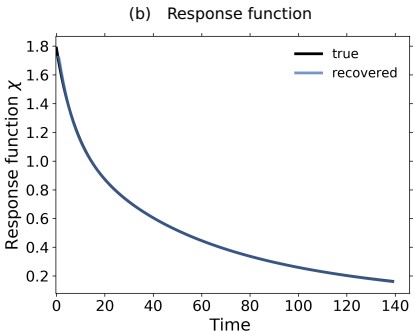

**Figure E5.** Spectrum $q_\lambda$ and response function $\chi(t)$ recovered from a 1% experiment performed with the toy model described in Part I taking an underlying discrete spectrum with three time scales $\tau = 7$, $\tau = 37$, $\tau = 100$. The data were taken with $SNR \sim \mathcal{O}(10^4) - \mathcal{O}(10^5)$.

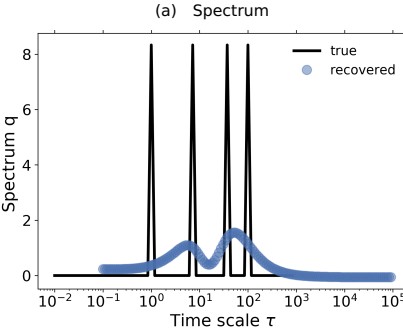
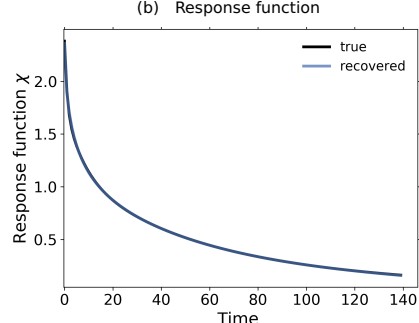

**Figure E6.** Spectrum $q_\lambda$ and response function $\chi(t)$ recovered from a 1% experiment performed with the toy model described in Part I taking an underlying discrete spectrum with four time scales $\tau = 1$, $\tau = 7$, $\tau = 37$, $\tau = 100$. The data were taken with $SNR \sim \mathcal{O}(10^4) - \mathcal{O}(10^5)$.

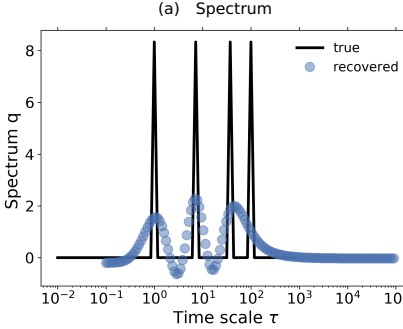
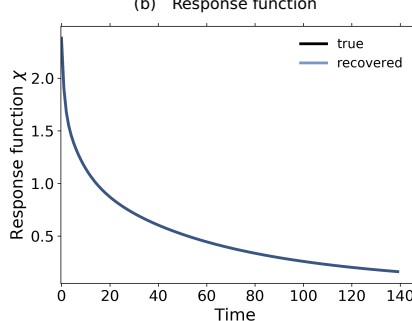

**Figure E7.** Spectrum $q_\lambda$ and response function $\chi(t)$ recovered from a $2 \times f_0$ experiment performed with the toy model described in Part I taking an underlying discrete spectrum with four time scales $\tau = 1$, $\tau = 7$, $\tau = 37$, $\tau = 100$. The data were taken with $SNR \sim \mathcal{O}(10^4) - \mathcal{O}(10^5)$.

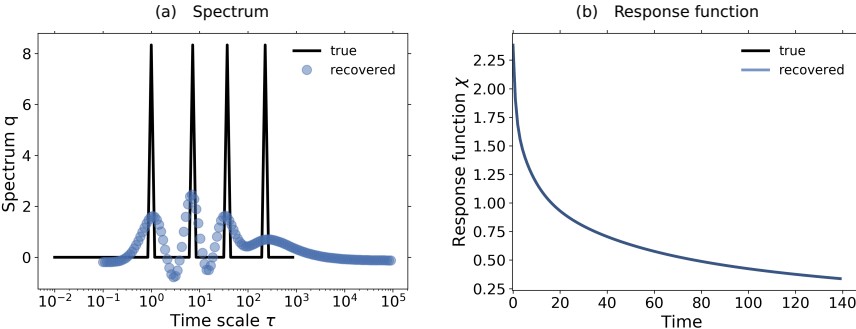

**Figure E8.** Spectrum $q_\lambda$ and response function $\chi(t)$ recovered from a $2 \times f_0$ experiment performed with the toy model described in Part I taking an underlying discrete spectrum with four time scales $\tau = 1$, $\tau = 7$, $\tau = 37$, $\tau = 228$. The data were taken with $SNR \sim \mathcal{O}(10^4) - \mathcal{O}(10^5)$.

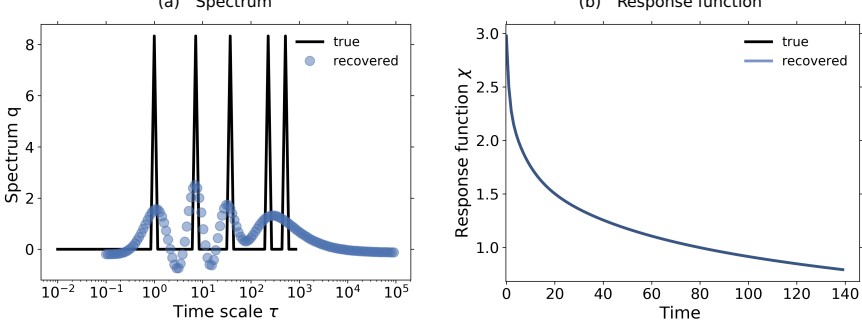

**Figure E9.** Spectrum $q_\lambda$ and response function $\chi(t)$ recovered from a $2 \times f_0$ experiment performed with the toy model described in Part I taking an underlying discrete spectrum with five time scales $\tau = 1$, $\tau = 7$, $\tau = 37$, $\tau = 228$, $\tau = 518$. The data were taken with $SNR \sim \mathcal{O}(10^4) - \mathcal{O}(10^5)$.





*Code and data availability.* The scripts employed to produce the results in this paper as well as information on how to obtain the underlying data can be found at http://hdl.handle.net/21.11116/0000-0008-0F06-2 (Torres Mendonca et al., 2021).

*Author contributions.* The ideas for this study were jointly developed by all authors. GTM conducted the study and wrote the first draft. All
authors contributed to the final manuscript.

*Competing interests.* The authors declare that they have no conflict of interest.

*Acknowledgements.* We would like to thank Andreas Chlond for his very helpful suggestions on the manuscript.





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
