# Peer review of "Identification of linear response functions from arbitrary perturbation experiments in the presence of noise Part II. Application to the land carbon cycle in the MPI Earth System Model"

_Nonlinear Processes in Geophysics, 2021_

## Author Comment (AC2)

**Response to Anonymous Referee #2 on "Identification of linear response functions from arbitrary perturbation experiments in the presence of noise**

**Part II. Application to the land carbon cycle in the MPI Earth System Model"**

Guilherme L. Torres Mendonça[1,2], Julia Pongratz[2,3], and Christian H. Reick[2]

[1]International Max Planck Research School on Earth System Modelling, Hamburg, Germany
[2]Max Planck Institute for Meteorology, Hamburg, Germany
[3]Ludwig-Maxmillians-Universität München, Munich, Germany

**Correspondence:** Guilherme L. Torres Mendonça (guilherme.mendonca@mpimet.mpg.de)

We would like to thank Anonymous Referee #2 for the positive review of our paper and for the useful suggestions. Below we give our point-by-point response to the comments.

**General comments**

*AR#2*: **(1) It would be useful to add a section to the introduction explaining why linear response functions are useful.**
5   **This is actually done in the outlook section, but it would be better placed in the introduction so readers understand why all of the dense mathematics is useful.**

*Authors*: We agree with the reviewer that indeed explaining in advance the applications of linear response functions would better motivate the study. Therefore we will move the applications mentioned in the outlook section to the introduction.

10  *AR#2*: **(2) Throughout the manuscript non-linearities are treated as a annoying problem to to overcome. However, Earth system models were build to study coupled climate carbon cycle feedbacks and to explore for the potential of non-linear behaviour in the Earth system. Thus the manuscript has a bit of a 'we linearized the system of equations and thereafter found everything to be linear' vibe. The manuscript need to be clearer about what non-linearities are and why they are important.**

15  *Authors*: We agree with the reviewer. In the revision we will better explain what we mean by nonlinearities, and what their relevance is.

*AR#2*: **(3) The manuscript has a bizarre way of referring to subplots as 'subfigure (x)' only giving the letter of the panel as x, without the figure number referred to. Please change these everywhere to the conventional Figure 1a, Fig 2b**

20 **ext.**

*Authors*: The text will be changed accordingly.

**Specific comments**

*AR#2*: **Line 10: The sentence "By taking instead of CO2 the resulting Net Primary Production as forcing, the response is approximately linear until CO2 perturbations of about 850 ppm." confusing, please rewrite.**

25 *AR#2*: **Line 32: Sentence is confusing.**

*Authors*: We will reformulate the referred sentences.

*AR#2*: **Line 33: Change 'is' to 'are'**

*Authors*: Agreed.

30

*AR#2*: **Line 50: You should include a full explanation in the introduction of what $\gamma$ and $\beta$ are. You have assumed the reader knows what they are. In many cases this will be true but including a full explanation means you will lose less people, the paper will be less intimidating and the technique will be more likely to be used.**

*Authors*: We thank the reviewer for the suggestion. The meaning of $\gamma$ and $\beta$ will be fully explained in the revision.

35

*AR#2*: **Line 61: The original C4MIP (Friedlingstein et al 2006) used a modified SRES scenario, not the 1pctCO2 experiment.**

*Authors*: This is true, but many other studies that followed Friedlingstein et al. (2006) adopted the 1pctCO2 experiment as standard for computing the climate-carbon sensitivities (e.g., Gregory et al., 2009; Arora et al., 2013; Schwinger et al., 2014;

40 Adloff et al., 2018; Williams et al., 2019; Arora et al., 2019). We will be more specific and mention these studies in the referred text.

*AR#2*: **Line 62: "performed with several" the 1pctCO2 experiment is part of DECK and is a required experiment for admittance to CMIP6.**

45 *Authors*: We will correct the text accordingly.

*AR#2*: **Line 65 to 69: Confusing long sentence. Break up for clarity.**

*Authors*: Agreed.

50 *AR#2*: **Line 81: Delete 'But while'. In general do not start sentences, let alone paragraphs, with 'but' in English. 'However' is acceptable.**

*Authors*: Agreed.

**AR#2: Line 113: Most of the audience will not know what 'ansatz' means.**

55 *Authors*: We will try to think of a better term.

**AR#2: Table 1: Clarify that % is compounded to cause and exponential rise in CO2 concentration.**

*Authors*: The clarification will be added in the revision.

60 **AR#2: Line 156: "cursed" is not the correct word to use here.**

*Authors*: We will find a more appropriate word.

**AR#2: Figure 2: Is the error metric non-dimensional? If so make this clear. 'Relative prediction error' may be clearer.**

*Authors*: Yes, it is. We will change the naming accordingly.

65

**AR#2: Figure 2: Be careful how you use the word 'forcing'. Many readers will assume radiative forcing unless this is specified otherwise.**

*Authors*: We will try to find a more appropriate term.

70 **AR#2: Equation 12: Lowercase 'c' is terrible notation for atmospheric CO2. $C_A$ or $C_{atm}$ would be clearer.**

*Authors*: The notation will be changed accordingly.

**AR#2: Line 291: Why use NPP instead of GPP. GPP is a direct measure of photosynthesis.**

*Authors*: The idea behind the splitting of the overall response of land carbon to $CO_2$ in two parts ($NPP(CO_2)$, $C_{land}(NPP)$)
75 is to separate off most of the nonlinearities into the the relation $NPP(CO_2)$ whose functional dependence can be guessed as logarithmic. Indeed, alternatively one could use GPP instead of NPP for the splitting, but in this way the nonlinearities arising from the dependence of autotrophic respiration on $CO_2$ would be part of the relation $C_{land}(GPP)$. Hence, using NPP instead of GPP is a better choice to split off most of the nonlinearities in $C_{land}(CO_2)$.

80 **AR#2: Line 295: "land carbon only via changes in photosyntetic productivity." This is not true. At higher atmospheric CO2 concentration plants are able to close their stoma more often, allowing higher retention of water. This decreases the rate of evapotranspiration, which feeds-back onto ground water supply and atmospheric processes depended of water vapour flux. This is well known and well studied phenomena.**

*Authors*: We agree with the reviewer's comment that $CO_2$ fertilization consists not only of a direct but also of an indirect effect:
85 rising $CO_2$ is affecting photosynthetic productivity directly via reduction of photorespiration in the biochemistry of the Calvin cycle, and indirectly by affecting stomatal closure, that via reduced transpiration feeds back on photosynthetic productivity by soil water savings that may e.g. lead to a prolonged growth period. But these two effects are taken into account by our sentence,

which says that "in the 'bgc' setup, perturbations in atmospheric $CO_2$ affect land carbon only via changes in photosynthetic productivity" – we never claimed that $CO_2$ acts only via its direct effect. On the other hand, it is true that the reduced transpiration may indeed lead to climatic changes, but these changes are small (e.g., Arora et al., 2013). To make these details clear, we will add a remark in the revision that in the bgc setup these two effects of $CO_2$ are active, and that in particular the indirect effect of stomatal closure may lead to small climatic changes.

*AR#2*: **432: This is not a Gregory plot, and calling it a Gregory plot is confusing. Do not refer to these are Gregory plots. Gregory plots are based on simple energy balance, these plots are not.**

*Authors*: We will think of a better name for the referred plots.

*AR#2*: **Line 569 to 570: This is not a complete sentence.**

*Authors*: We will reformulate the sentence.

With best regards,

Guilherme L. Torres Mendonça, Julia Pongratz and Christian H. Reick

**References**

Adloff, M., Reick, C. H., and Claussen, M.: Earth system model simulations show different feedback strengths of the terrestrial carbon cycle under glacial and interglacial conditions, Earth System Dynamics, 9, 413–425, 2018.

Arora, V. K., Boer, G. J., Friedlingstein, P., Eby, M., Jones, C. D., Christian, J. R., Bonan, G., Bopp, L., Brovkin, V., Cadule, P., et al.: Carbon–concentration and carbon–climate feedbacks in CMIP5 Earth system models, Journal of Climate, 26, 5289–5314, 2013.

Arora, V. K., Katavouta, A., Williams, R. G., Jones, C. D., Brovkin, V., Friedlingstein, P., Schwinger, J., Bopp, L., Boucher, O., Cadule, P., et al.: Carbon-concentration and carbon-climate feedbacks in CMIP6 models, and their comparison to CMIP5 models, Biogeosciences Discussions, pp. 1–124, 2019.

Friedlingstein, P., Cox, P., Betts, R., Bopp, L., von Bloh, W., Brovkin, V., Cadule, P., Doney, S., Eby, M., Fung, I., et al.: Climate–carbon cycle feedback analysis: results from the C4MIP model intercomparison, Journal of climate, 19, 3337–3353, 2006.

Gregory, J. M., Jones, C., Cadule, P., and Friedlingstein, P.: Quantifying carbon cycle feedbacks, Journal of Climate, 22, 5232–5250, 2009.

Schwinger, J., Tjiputra, J. F., Heinze, C., Bopp, L., Christian, J. R., Gehlen, M., Ilyina, T., Jones, C. D., Salas-Mélia, D., Segschneider, J., et al.: Nonlinearity of ocean carbon cycle feedbacks in CMIP5 earth system models, Journal of Climate, 27, 3869–3888, 2014.

Williams, R. G., Katavouta, A., and Goodwin, P.: Carbon-cycle feedbacks operating in the climate system, Current Climate Change Reports, 5, 282–295, 2019.

---

## Author Response (AR1)

**Revision of the manuscript "Identification of linear response functions from arbitrary perturbation experiments in the presence of noise**

**Part II. Application to the land carbon cycle in the MPI Earth System Model"**

Guilherme L. Torres Mendonça[1,2], Julia Pongratz[2,3], and Christian H. Reick[2]

[1]International Max Planck Research School on Earth System Modelling, Hamburg, Germany
[2]Max Planck Institute for Meteorology, Hamburg, Germany
[3]Ludwig-Maxmillians-Universität München, Munich, Germany

**Correspondence:** Guilherme L. Torres Mendonça (guilherme.mendonca@mpimet.mpg.de)

Dear Editor, dear Reviewers,

We would like to thank you for your valuable suggestions. Below we provide:

1. Comments from the referees and our responses, as posted in the online discussion (in black).

2. Description of the corresponding changes in the manuscript (in blue)*.

3. Manuscript with all revisions highlighted.

In a separate file you find as well the final revised paper.

With best regards,

10    Guilherme L. Torres Mendonça, Julia Pongratz and Christian H. Reick

Hamburg, June 30, 2021

*When describing the changes in the manuscript, please note that the lines we refer to are those in the manuscript *with the highlighted revisions below* (and not those in the final revised paper attached separately).

**Response to Anonymous Referee #1**

We thank Anonymous Referee #1 for the positive review of our paper. Below we respond to the comments point by point.

*AR#1*: **Line 193 - Is it too strong a statement to say that the prediction error comes solely from the deterioration of the recovery of the response function and not from nonlinearities? The optimal (or even appropriate) form and coefficients in subgrid parameterisations of any one of the many unresolved processes within Earth system models would in general be dependent on the level of forcing.**

*Authors*: As mentioned in the introduction of Part I, the linear response equation (1) can be seen as the linear term in a Volterra series, which is an expansion of the response of the system into the perturbation that under certain assumptions completely describes this response for any time-dependent perturbation. In this expansion, all nonlinearities of the system, *regardless of their origin*, are represented by the terms nonlinear in the forcing. Therefore, as long as the forcing strength is sufficiently small, such nonlinearities do not influence the response, which can be completely described by Eq. (1) for any time-dependent forcing once the linear response function is known.

*Changes*: Also reviewer #2 asked us to make clearer what we mean when talking of nonlinearities. Such explanation has been added (L195–210 in the manuscript with highlighted revisions).

*AR#1*: **Could this type of nonlinearity influence the deterioration in the recovery of the response functions? Note, in general parameterisations developed in GCMs are designed / tuned to reproduce the historically observed climate, and not necessarily future ones with different forcing.**

*Authors*: Yes, for large enough forcing strength such nonlinearities from parametrizations do deteriorate the recovery of the response function. In fact, an example where this happens is shown in Fig. 6(a): There, one sees that when the response function $\chi_\beta(t)$ is derived from data taken for forcing strengths larger than about 100 ppm, the prediction error starts to increase. As explained in section 2.2, this increase in the prediction error can be unambiguously traced back to the deterioration in the recovery of the response function caused by nonlinearities in the 1%-experiment data from which the response function was derived. As we found out in the study, the origin of this deterioration is the nonlinear relationship between NPP and atmospheric $CO_2$.

*Changes*: No action taken.

*AR#1*: **To what extent are the responses sensitive to the historical trajectory of the emissions? For example could the sensitivities determine in the the 1% runs be used to determine the responses in a more temporally complex trajectory as specified in the ScenarioMIP project?**

*Authors*: As explained above – and also in the introduction of Part I –, in principle the derived response functions can be employed to predict the response to any time-dependent perturbation, as long as the perturbation strength is sufficiently small.

Therefore, the temporal complexity of the perturbation trajectory would not be an issue. Nevertheless, the response functions

that we derive in the present study describe ultimately the response of the land carbon cycle to perturbations in atmospheric $CO_2$. In historical simulations one is not in the ideal situation where the only perturbation to the system is the increase in atmospheric $CO_2$: There is also land-use change, other greenhouse gases, etc. Therefore, studying historical simulations by means of linear response functions is in principle possible, but one would have to take into account the response to these additional perturbations. This is indeed an interesting research direction, but it sidetracks from the core of our study. Therefore we would prefer not to address this comment in the revised manuscript.

*Changes*: No action taken.

*AR#1*: **To what extent are the responses sensitive to the spatial distribution of the emissions? Clearly there are many spatial distributions that could produce the same global average (or even tropical / extra-tropical averages). Presumably not all of these distributions would produce the same response.**

*Authors*: Indeed the global response could be different if the different regions would have a different $CO_2$ concentration. In such a case the reviewer is right: If the $CO_2$ was very different in different regions, then the tropical/extra-tropical response functions we derived would not show what is really happening in these regions. In this case, we would have to derive the functions taking into account the different $CO_2$ concentrations for the different regions – as we did when deriving $\chi_{NPP}(t)$ for different NPP perturbations over tropics and extra-tropics (Eqs. (19) and (20)). But since in reality $CO_2$ is so well mixed, this is a hypothetical consideration. We will clarify in the manuscript that the tropical/extra-tropical response functions $\chi_{\beta}^{tr}(t)$ and $\chi_{\beta}^{et}(t)$ can be interpreted as the locally correct response functions because of $CO_2$ being well mixed.

*Changes*: The clarification was added to the text (L533–535 in the manuscript with highlighted revisions).

*AR#1*: **Following this line of thinking, could one consider the response functions between the global average quantities as the first scale in a spatial spectrum? Could one calculate response functions per say the principal components (PCs) of land carbon and PCs of surface temperature, or some other modal decomposition of these fields?**

*Authors*: Yes, in principle these functions can also be employed to study the dynamics of the system spatially. We have shown a simple example of this by looking at the responses for tropics and extra-tropics, but in the literature one finds also applications to study latitudinal distributions and even entire global maps (Thompson and Randerson, 1999; Lucarini et al., 2017). It would therefore indeed be an interesting idea to employ these functions to investigate the spatial response of the carbon cycle as suggested by the referee. Nevertheless, this idea is also beyond the scope of the present study, and therefore we would prefer not to address it in the revisions.

*Changes*: No action taken.

**Response to Anonymous Referee #2**

We would like to thank Anonymous Referee #2 for the positive review of our paper and for the useful suggestions. Below we give our point-by-point response to the comments.

**General comments**

*AR#2*: **(1) It would be useful to add a section to the introduction explaining why linear response functions are useful. This is actually done in the outlook section, but it would be better placed in the introduction so readers understand why all of the dense mathematics is useful.**

*Authors*: We agree with the reviewer that indeed explaining in advance the applications of linear response functions would better motivate the study. Therefore we will move the applications mentioned in the outlook section to the introduction.

*Changes*: Trying to better motivate the study by moving the application examples from the outlook to the introduction unfortunately did not work out because these examples are too specific. Therefore we decided to keep the outlook as it is, but add to the introduction examples demonstrating typical applications of response functions as known from the literature (L36–44 in the manuscript with highlighted revisions). Moreover we added some general remarks on the potential of linear response function for analyzing processes at different time scales, a field of research just emerging (L45–75). We hope that in this way the reader is now better motivated to follow us through the mathematical parts of the study.

*AR#2*: **(2) Throughout the manuscript non-linearities are treated as a annoying problem to to overcome. However, Earth system models were build to study coupled climate carbon cycle feedbacks and to explore for the potential of non-linear behaviour in the Earth system. Thus the manuscript has a bit of a 'we linearized the system of equations and thereafter found everything to be linear' vibe. The manuscript need to be clearer about what non-linearities are and why they are important.**

*Authors*: We agree with the reviewer. In the revision we will better explain what we mean by nonlinearities, and what their relevance is.

*Changes*: Done (L195–210).

*AR#2*: **(3) The manuscript has a bizarre way of referring to subplots as 'subfigure (x)' only giving the letter of the panel as x, without the figure number referred to. Please change these everywhere to the conventional Figure 1a, Fig 2b ext.**

*Authors*: The text will be changed accordingly.

*Changes*: Done (throughout the paper; see manuscript with revisions).

**Specific comments**

*AR#2*: **Line 10: The sentence "By taking instead of CO2 the resulting Net Primary Production as forcing, the response is approximately linear until CO2 perturbations of about 850 ppm." confusing, please rewrite.**

*Changes*: Done (L10–11).

*AR#2*: **Line 32: Sentence is confusing.**

*Authors*: We will reformulate the referred sentences.

*Changes*: Done (L31–33).

*AR#2*: **Line 33: Change 'is' to 'are'**

*Authors*: Agreed.

*Changes*: We decided to keep "is" to be consistent with our use of dynamics in the singular in other parts of the paper. According-ing to the Merriam-Webster dictionary, "dynamics" is plural in form but singular *or* plural in construction (https://www.merriam-webster.com/dictionary/dynamics).

*AR#2*: **Line 50: You should include a full explanation in the introduction of what $\gamma$ and $\beta$ are. You have assumed the reader knows what they are. In many cases this will be true but including a full explanation means you will lose less people, the paper will be less intimidating and the technique will be more likely to be used.**

*Authors*: We thank the reviewer for the suggestion. The meaning of $\gamma$ and $\beta$ will be fully explained in the revision.

*Changes*: Done (L70–82).

*AR#2*: **Line 61: The original C4MIP (Friedlingstein et al 2006) used a modified SRES scenario, not the 1pctCO2 experiment.**

*Authors*: This is true, but many other studies that followed Friedlingstein et al. (2006) adopted the 1pctCO2 experiment as standard for computing the climate-carbon sensitivities (e.g., Gregory et al., 2009; Arora et al., 2013; Schwinger et al., 2014; Adloff et al., 2018; Williams et al., 2019; Arora et al., 2019). We will be more specific and mention these studies in the referred text.

*Changes*: We have included the referred studies in the text (L94–95).

*AR#2*: **Line 62: "performed with several" the 1pctCO2 experiment is part of DECK and is a required experiment for admittance to CMIP6.**

*Authors*: We will correct the text accordingly.

*Changes*: The text was corrected (L95).

*AR#2*: **Line 65 to 69: Confusing long sentence. Break up for clarity.**

*Authors*: Agreed.

*Changes*: Done (L98–99).

155    *AR#2*: **Line 81: Delete 'But while'. In general do not start sentences, let alone paragraphs, with 'but' in English. 'However' is acceptable.**

*Authors*: Agreed.

*Changes*: "But" was substituted by "However" (L114).

160    *AR#2*: **Line 113: Most of the audience will not know what 'ansatz' means.**

*Authors*: We will try to think of a better term.

*Changes*: We now use the word "relation" (L146–147).

*AR#2*: **Table 1: Clarify that % is compounded to cause and exponential rise in CO2 concentration.**

165    *Authors*: The clarification will be added in the revision.

*Changes*: Done (see Table 1).

*AR#2*: **Line 156: "cursed" is not the correct word to use here.**

*Authors*: We will find a more appropriate word.

170    *Changes*: We changed "cursed" to "complicated" (L189).

*AR#2*: **Figure 2: Is the error metric non-dimensional? If so make this clear. 'Relative prediction error' may be clearer.**

*Authors*: Yes, it is. We will change the naming accordingly.

*Changes*: The naming was changed in Figs. 2, 3, 6, 7 and also in the text (see highlighted revisions).

175

*AR#2*: **Figure 2: Be careful how you use the word 'forcing'. Many readers will assume radiative forcing unless this is specified otherwise.**

*Authors*: We will try to find a more appropriate term.

*Changes*: We preferred to keep the term "forcing". In this particular figure we are plotting results from a purely mathematical
180    toy model so that the forcing has no physical meaning. In the application to the land carbon confusion with radiative forcing is not expected because we indicate also the units of the forcing in the axis label (Figs. 3 and 6).

*AR#2*: **Equation 12: Lowercase 'c' is terrible notation for atmospheric CO2. $C_A$ or $C_{atm}$ would be clearer.**

*Authors*: The notation will be changed accordingly.

185    *Changes*: The notation was changed throughout the text and in Fig. 7.

*AR#2*: **Line 291: Why use NPP instead of GPP. GPP is a direct measure of photosynthesis.**

*Authors*: The idea behind the splitting of the overall response of land carbon to $CO_2$ in two parts ($NPP(CO_2)$, $C_{land}(NPP)$) is to separate off most of the nonlinearities into the the relation $NPP(CO_2)$ whose functional dependence can be guessed as logarithmic. Indeed, alternatively one could use GPP instead of NPP for the splitting, but in this way the nonlinearities arising from the dependence of autotrophic respiration on $CO_2$ would be part of the relation $C_{land}(GPP)$. Hence, using NPP instead of GPP is a better choice to split off most of the nonlinearities in $C_{land}(CO_2)$.

*Changes*: No action taken.

*AR#2*: **Line 295: "land carbon only via changes in photosyntetic productivity." This is not true. At higher atmospheric CO2 concentration plants are able to close their stoma more often, allowing higher retention of water. This decreases the rate of evapotranspiration, which feeds-back onto ground water supply and atmospheric processes depended of water vapour flux. This is well known and well studied phenomena.**

*Authors*: We agree with the reviewer's comment that $CO_2$ fertilization consists not only of a direct but also of an indirect effect: rising $CO_2$ is affecting photosynthetic productivity directly via reduction of photorespiration in the biochemistry of the Calvin cycle, and indirectly by affecting stomatal closure, that via reduced transpiration feeds back on photosynthetic productivity by soil water savings that may e.g. lead to a prolonged growth period. But these two effects are taken into account by our sentence, which says that "in the 'bgc' setup, perturbations in atmospheric $CO_2$ affect land carbon only via changes in photosynthetic productivity" – we never claimed that $CO_2$ acts only via its direct effect. On the other hand, it is true that the reduced transpiration may indeed lead to climatic changes, but these changes are small (e.g., Arora et al., 2013). To make these details clear, we will add a remark in the revision that in the bgc setup these two effects of $CO_2$ are active, and that in particular the indirect effect of stomatal closure may lead to small climatic changes.

*Changes*: A remark was included accordingly (see footnote 1, L349).

*AR#2*: **432: This is not a Gregory plot, and calling it a Gregory plot is confusing. Do not refer to these are Gregory plots. Gregory plots are based on simple energy balance, these plots are not.**

*Authors*: We will think of a better name for the referred plots.

*Changes*: Gregory's great idea was to plot the relaxation of fluxes towards equilibrium after a step function perturbation to gain insight into inherent properties of the system – in the case of radiative forcing analyzed by Gregory the climate sensitivity, in our case the internal time scales. This is a genuine idea of Gregory that we follow here and therefore should give credits to him. We thus decided to name our plots "Gregory-type plots" instead of "Gregory plots". In conjunction with some additional remarks for this naming we thus hope to prevent any confusion (L486–490, L500–502, L560, L667).

*AR#2*: **Line 569 to 570: This is not a complete sentence.**

*Authors*: We will reformulate the sentence.

*Changes*: The sentence was reformulated accordingly (L630).

[revised manuscript text omitted]

$$\chi^{tr}_\beta(t) = \frac{\partial NPP^{tr}}{\partial c}\frac{\partial NPP^{tr}}{\partial C_{atm}}\bigg|_{c=c_{PI}, C_{atm}=C^0_{atm}} \chi^{tr}_{NPP}(t),\tag{27}$$

$$\chi^{et}_\beta(t) = \frac{\partial NPP^{et}}{\partial c}\frac{\partial NPP^{et}}{\partial C_{atm}}\bigg|_{c=c_{PI}, C_{atm}=C^0_{atm}} \chi^{et}_{NPP}(t).\tag{28}$$

[Figure]

**Figure 10.** Investigation of the land carbon response in the tropics and extra-tropics and how the regional response functions combine to the global response functions. The analysis is based on the $2\times CO_2$ bgc experiment. (a) Laplace transform $\widetilde{\chi}_{NPP}(p)$ of global $\chi_{NPP}(t)$ obtained directly from the global carbon response and from combining the tropical and extra-tropical response functions; (b) $\chi_\beta(t)$ obtained directly from the global carbon response and from combining the tropical and extra-tropical response functions; (c) As (b) but for $q_\beta(\tau)$; (d) Decomposition of $q_\beta(\tau)$ into tropical and extra-tropical spectra (Eq. (31)). In (c) and (d) the dots and crosses indicate the computed values, while the connecting lines are only inserted to guide the eye. For more details see text.

 Note that because $CO_2$ is well mixed, $\Delta C_{atm}^{tr}(t) \approx \Delta C_{atm}^{et}(t) \approx \Delta C_{atm}(t)$, so that, by characterizing the tropical and the extra-tropical response to $CO_2$, $\chi_\beta^{tr}(t)$ and $\chi_\beta^{et}(t)$ are describing the response to a regionally correct perturbation. Using the response functions obtained from Eqs. (27) and (28), one can now write Eq. (14) for global,

535

tropical and extra-tropical carbon. Plugging the result into Eq. (24) gives

$$\int_0^t \chi_\beta(t-s)\Delta \underline{c C_{atm}}(s)ds = \int_0^t [\chi_\beta^{tr}(t-s) + \chi_\beta^{et}(t-s)]\Delta \underline{c C_{atm}}(s)ds. \tag{29}$$

Hence, one can infer that

$$\chi_\beta(t) = \chi_\beta^{tr}(t) + \chi_\beta^{et}(t). \tag{30}$$

540    Therefore, the global response function $\chi_\beta(t)$ can be obtained from $\chi_\beta^{tr}(t)$ and $\chi_\beta^{et}(t)$. But in addition, since $\chi(t)$ is given by Eq. (5), Eq. (30) implies that one can also obtain the global spectrum $q_\beta(\tau)$ by combining the regional spectra:

$$q_\beta(\tau) = q_\beta^{tr}(\tau) + q_\beta^{et}(\tau). \tag{31}$$

Therefore, using the recovered response functions for tropical and extra-tropical carbon one can obtain the global response function $\chi_\beta$ and its associated spectrum $q_\beta$. Accordingly, in this test we check Eqs. (30) and (31). For the calculation of the
545    derivatives in Eqs. (27) and (28) we fitted $\sout{NPP^{tr} = NPP^{tr}(c) \text{ and } NPP^{et} = NPP^{et}(c)}$ $\underline{NPP^{tr} = NPP^{tr}(C_{atm})}$ and $\underline{NPP^{et} = NPP^{et}(C_{atm})}$ once again by a polynomial of order 6 (which obtained the best results for global NPP in Fig. 7(b)) and took the derivatives from the fits. For $\chi_\beta(t)$ we used the recovery with best quality from Fig. 7 ("NPP$_{pol6}^{2\times}$"). The spectra $q_\beta(\tau)$, $q_\beta^{tr}(\tau)$ and $q_\beta^{et}(\tau)$ are obtained by scaling the spectra from $\chi_{NPP}(t)$, $\chi_{NPP}^{tr}(t)$ and $\chi_{NPP}^{et}(t)$ by the respective derivatives $\left.\frac{\partial NPP}{\partial c}\right|_{c=c_{PI}}$, $\left.\frac{\partial NPP^{tr}}{\partial c}\right|_{c=c_{PI}}$ $\sout{\text{
[revised manuscript text omitted]

---

## Author Response (AR2)

**Revision of the manuscript "Identification of linear response** functions from arbitrary perturbation experiments in the presence of noise**

**Part II. Application to the land carbon cycle in the MPI Earth System Model"**

Guilherme L. Torres Mendonça1,2, Julia Pongratz2,3, and Christian H. Reick2

1International Max Planck Research School on Earth System Modelling, Hamburg, Germany 2Max Planck Institute for Meteorology, Hamburg, Germany 3Ludwig-Maxmillians-Universität München, Munich, Germany

Correspondence: Guilherme L. Torres Mendonça (guilherme.mendonca@mpimet.mpg.de)

Dear Editor,

We would like to thank you and the reviewers for once more considering our paper. You pointed out that four of our responses to the first round of reviews did not lead to corresponding changes in the manuscript, and therefore suggested to

5 introduce such changes in the text. We have now added these revisions, and also rightfully thanked the anonymous referees for their contributions to our paper.

Below we report our revisions by

- 1. Presenting those referee comments for which previously no changes were added to the manuscript (in black).
- 2. Describing the new manuscript changes addressing those comments (in blue)\*.
- 10 3. Appending the reviewed manuscript with all revisions highlighted.

In a separate file you find as well the final revised paper.

With best regards,

Guilherme L. Torres Mendonça, Julia Pongratz and Christian H. Reick

15

Hamburg, August 19, 2021

\*When describing the changes in the manuscript, please note that the lines we refer to are those in the manuscript *with the* 20 *highlighted revisions below* (and not those in the final revised paper attached separately).

**Response to Anonymous Referee #1**

40

AR#1: Could this type of nonlinearity influence the deterioration in the recovery of the response functions? Note, in
 general parameterisations developed in GCMs are designed / tuned to reproduce the historically observed climate, and not necessarily future ones with different forcing.

Authors: Yes, for large enough forcing strength such nonlinearities from parametrizations do deteriorate the recovery of the response function. In fact, an example where this happens is shown in Fig. 6(a): There, one sees that when the response function  $\chi_{\beta}(t)$  is derived from data taken for forcing strengths larger than about 100 ppm, the prediction error starts to increase.

30 As explained in section 2.2, this increase in the prediction error can be unambiguously traced back to the deterioration in the recovery of the response function caused by nonlinearities in the 1%-experiment data from which the response function was derived. As we found out in the study, the origin of this deterioration is the nonlinear relationship between NPP and atmospheric CO2.

Changes: We have now changed a few sentences in the discussions section to make it more explicit that the recovery of the

35 response function  $\chi_{\beta}$  deteriorates because of nonlinearities in the biogeochemical response of the land carbon (see L654–661 in the manuscript with highlighted revisions).

*AR#1*: To what extent are the responses sensitive to the historical trajectory of the emissions? For example could the sensitivities determine in the the 1% runs be used to determine the responses in a more temporally complex trajectory as specified in the ScenarioMIP project?

- *Authors*: As explained above and also in the introduction of Part I –, in principle the derived response functions can be employed to predict the response to any time-dependent perturbation, as long as the perturbation strength is sufficiently small. Therefore, the temporal complexity of the perturbation trajectory would not be an issue. Nevertheless, the response functions that we derive in the present study describe ultimately the response of the land carbon cycle to perturbations in atmospheric
- 45 CO2. In historical simulations one is not in the ideal situation where the only perturbation to the system is the increase in atmospheric CO2: There is also land-use change, other greenhouse gases, etc. Therefore, studying historical simulations by means of linear response functions is in principle possible, but one would have to take into account the response to these additional perturbations. This is indeed an interesting research direction, but it sidetracks from the core of our study. Therefore we would prefer not to address this comment in the revised manuscript.
- 50 *Changes*: We have added a paragraph to the outlook where we mention that our analysis may in principle be extended to more realistic situations such as historical simulations (see L731–735).

*AR#1*: Following this line of thinking, could one consider the response functions between the global average quantities as the first scale in a spatial spectrum? Could one calculate response functions per say the principal components

55 (PCs) of land carbon and PCs of surface temperature, or some other modal decomposition of these fields?

*Authors*: Yes, in principle these functions can also be employed to study the dynamics of the system spatially. We have shown a simple example of this by looking at the responses for tropics and extra-tropics, but in the literature one finds also applications to study latitudinal distributions and even entire global maps (Thompson and Randerson, 1999; Lucarini et al., 2017). It would therefore indeed be an interesting idea to employ these functions to investigate the spatial response of the carbon cycle as

60

suggested by the referee. Nevertheless, this idea is also beyond the scope of the present study, and therefore we would prefer not to address it in the revisions.

*Changes*: We have introduced our response in the additional paragraph in the outlook section to make it clear that such an extension of our analysis to study the spatial response of the carbon cycle is also possible (L735–738).

**65 **Response to Anonymous Referee #2**

75

**AR#2: Line 291: Why use NPP instead of GPP. GPP is a direct measure of photosynthesis.**

Authors: The idea behind the splitting of the overall response of land carbon to  $CO_2$  in two parts (NPP(CO2),  $C_{land}$ (NPP))

70 is to separate off most of the nonlinearities into the the relation NPP(CO2) whose functional dependence can be guessed as logarithmic. Indeed, alternatively one could use GPP instead of NPP for the splitting, but in this way the nonlinearities arising from the dependence of autotrophic respiration on CO2 would be part of the relation  $C_{land}$ (GPP). Hence, using NPP instead of GPP is a better choice to split off most of the nonlinearities in  $C_{land}$ (CO2).

*Changes*: We have now explicitly explained why we take NPP instead of GPP in the text (see L338–364 in the manuscript with highlighted revisions).

**References**

[revised manuscript text omitted]